

# Climatic niche evolution in the viviparous *Sceloporus torquatus* group (Squamata: Phrynosomatidae)

Norberto Martínez-Méndez[1], Omar Mejía[2], Jorge Ortega[1] and Fausto Méndez-de la Cruz[3]

[1] Departamento de Zoología, Laboratorio de Bioconservación y Manejo, Escuela Nacional de Ciencias Biológicas del Instituto Politécnico Nacional, Ciudad de México, México
[2] Departamento de Zoología, Laboratorio de Variación y Evolución, Escuela Nacional de Ciencias Biológicas del Instituto Politécnico Nacional, Ciudad de México, México
[3] Departamento de Zoología, Laboratorio de Herpetología, Instituto de Biología de la Universidad Nacional Autónoma de México, Ciudad de México, México

Corresponding author
Norberto Martínez-Méndez,
nomartinezm@ipn.mx

## ABSTRACT

The cold-climate hypothesis maintains that viviparity arose as a means to prevent increased egg mortality in nests owing to low temperatures, and this hypothesis represents the primary and most strongly supported explanation for the evolution of viviparity in reptiles. In this regard, certain authors have stated that viviparous species will exhibit speciation via climatic niche conservatism, with similar climatic niches being observed in allopatric sister species. However, this prediction remains to be tested with bioclimatic variables relevant to each viviparous group. In the present study, we examined climatic niche evolution in a group of North American viviparous lizards to determine whether their diversification is linked to phylogenetic niche conservatism (PNC). We evaluated the phylogenetic signal and trait evolution of individual bioclimatic variables and principal component (PC) scores of a PC analysis, along with reconstructions of ancestral climate tolerances. The results suggest that diversification of the *Sceloporus torquatus* group species is associated with both niche differentiation and PNC. Furthermore, we did not observe PNC across nearly all bioclimatic variables and in PC2 and PC3. However, in Precipitation Seasonality (Bio15), in Precipitation of Coldest Quarter (Bio19) and in PC1 (weakly associated with variability of temperature), we did observe PNC. Additionally, variation of the scores along the phylogeny and Pagel's delta ($\delta$) >1 of PC3 suggests a fast, recent evolution to dry conditions in the clade that sustains *S. serrifer*.

## BACKGROUND

Viviparity among squamate reptiles (lizards and snakes) has evolved from oviparity approximately 100 times (*Blackburn, 2000*, *2015*) and has been a model system for testing many evolutionary hypotheses regarding the origin of viviparity in vertebrates

(*Lambert & Wiens, 2013*). In this regard, viviparity among reptiles has been linked to cold climates as it provides a selective advantage that prevents the death of embryos in nests owing to low temperatures (*Tinkle & Gibbons, 1977*; *Shine, 1985*; *Lambert & Wiens, 2013*; *Ma et al., 2018*). Moreover, evidence suggests that certain lizard species that have evolved viviparity remain adaptively constrained to cold climates (*Pincheira-Donoso et al., 2013*). However, few viviparous lizard species of the genus *Sceloporus* secondarily invaded warm climates (*Lambert & Wiens, 2013*). Within lizards of the family Phrynosomatidae (*Sceloporus* and *Phrynosoma*), viviparity has been demonstrated to evolve more often in tropical montane regions than temperate regions, which is explained by the presence of high-elevation and cool climate specialists based on greater seasonal temperature stability at tropical latitudes (*Lambert & Wiens, 2013*). Simultaneously, the tropical montane species are more likely to be isolated on mountaintops because of the climatic zonation of tropical regions. Consequently, it is expected that viviparous species will exhibit speciation via climatic niche conservatism with similar climatic niches being observed among the allopatric sisters species (*Lambert & Wiens, 2013*). Notably, this prediction remains to be explored not only through the analysis of climatic variables linked with thermal niche, but also a wider set of bioclimatic variables.

Climatic niche conservatism, or, more precisely, phylogenetic niche conservatism (PNC), is the tendency of related species to retain their ancestral requirements or niche throughout time (*Boucher et al., 2014*). PNC has commonly been studied by measuring phylogenetic signal (PS). PS is the tendency for related species to resemble each other more than they resemble species drawn at random from a phylogenetic tree (*Blomberg & Garland, 2002*), and some authors consider a PS signal sufficient for verification of PNC (*Wiens et al., 2010b*). However, a number of revisions have highlighted theoretical problems with the PNC concept, as well as practical difficulties related to its measurement (*Revell, Harmon & Collar, 2008*; *Münkemüller et al., 2015*). In fact, various authors argue that PNC is a process, while others consider it a pattern and others still argue that PNC can be either a process or a pattern depending on how the research questions are developed (*Losos, 2008*; *Wiens et al., 2010b*). In this regard, *Losos (2008)* pointed out that PNC could result from several processes (i.e., genetic constraints or stabilizing selection), notwithstanding some comparative approaches being useful for exploring or rejecting processes.

However, the importance of PNC studies lie in a combination or interaction between niche evolution and niche conservatism shaping the biogeographic patterns and distributions observed in many species (*Wiens & Donoghue, 2004*) as well as the functional diversification of lineages and niche similarity of phylogenetically related species (*Culumber & Tobler, 2016*). In this sense, the current geographic distribution of species can be explained as the interaction of historical processes studied by biogeography, such as vicariance and dispersal, along with shallow time processes that include ecological factors, including habitat filtering, biotic variables such as competition or predation, and niche partitioning (*Sexton et al., 2009*; *Nyári & Reddy, 2013*). Depending on the author, the ecological niche of a species could refer to:

(1) Hutchinsonian niche, which includes biotic and abiotic variables allowing the persistence of populations (*Hutchinson, 1957*); (2) Grinellian niche, which focuses on the environmental space of non-interacting and non-linked abiotic variables where the species survive and reproduce (*Grinnell, 1917*); or (3) Eltonian niche, which refers to the functional role of the species (*Elton, 1927*).

Abiotic variables are important for the speciation process—reproductive isolation could appear by the evolution of barriers to gene flow owing to divergent natural selection (*Mayr, 1947*; *Pavey et al., 2010*; *Nosil, 2012*), though many authors pointed out that sexual selection is also required to complete the speciation process (*Maan & Seehausen, 2011*; *Servedio & Boughman, 2017*). Ecologically mediated speciation implies changes in the ecological niche; however, ecological niches are multidimensional, and it is thus unlikely that each dimension evolves in the same manner (*Schluter, 1996*; *Ackerly, 2003*; *Duran, Meyer & Pie, 2013*). Cases have been observed wherein reproductive isolation is conditioned by a combination of ecological constraints and a vicariance process (e.g., geographic barriers), where species could retain certain ancestral requirements that limit adaptation to climatic conditions imposed by the barrier (*Wiens & Graham, 2005*).

One particular group of lizards is particularly suitable for the study of Grinellian niche evolution of viviparous species—the genus *Scelopurus*. *Sceloporus* species are widely distributed in North America, and the genus contains approximately 70 viviparous species distributed across five groups (*Wiens & Reeder, 1997*; *Méndez-De La Cruz, Villagrán-Santa Cruz & Andrews, 1998*), and molecular and phylogenetic information is available for nearly all recognized species along with a comprehensive occurrence database (*Wiens & Reeder, 1997*; *Leaché, 2010*; *Wiens et al., 2010a*; *Leaché et al., 2016*). Among viviparous groups of the genus *Sceloporus*, the *torquatus* group (*Smith, 1938*) has interesting characteristics for a phyloclimatic study because the climatic gradient where this population is found. The *torquatus* group has a wide distribution—from the southern US southward into Guatemala (*Martínez-Méndez & Méndez De La Cruz, 2007*)—and the group is found in mountain ranges with temperate conditions throughout its distribution, with a few species occur in semi-desert and tropical lowland environments (e.g., *Sceloporus serrifer*).

Given the hypothesis that lizard viviparous species in tropical latitudes exhibit speciation via climatic niche conservatism and similar niches in allopatric sister species, we used the viviparous *S. torquatus* group as a model to: (1) assess if niche evolution among species of the group is consistent with PNC; and (2) test whether niche overlap values and environmental tolerances among species relative to their phylogenetic relationships are similar among sister species.

In order to achieve the stated objectives, we constructed a phylogeny of the group and employed a phyloclimatic analysis using occurrence data and bioclimatic variables to: (1) evaluate the PS of bioclimatic variables comprising species Grinellian niches; (2) fit macroevolutionary models for the bioclimatic variables used; (3) investigate the history of ecological niche occupancy and accumulation; (4) assess ancestral climatic tolerances; and (5) calculate ecological niche disparity through time (DTT).

## MATERIALS AND METHODS

### Data sources

Occurrence data were obtained from The Global Biodiversity Information Facility (http://www.gbif.org/), HERPNET (http://www.herpnet.org), Comisión Nacional para el Conocimiento y Uso de la Biodiversidad (https://www.gob.mx/conabio), and from the field notes of the main author. We removed occurrence records that constituted coordinate errors (i.e., points on the sea) and similar coordinates. To minimize spatial autocorrelation, we randomly removed occurrences within 0.5 km of each other in order to obtain localities in distinct grids to match the spatial resolution of environmental layers (30 arc seconds). For environmental layers, we used bioclim layers at a 30 arc second resolution (approximately $1 \times 1$ km) including monthly and annual maximum and minimum temperature and precipitation variables from the WorldClim Database 1.4 (http://www.worldclim.org). Additionally, we used monthly and annual potential evapotranspiration (PET) and aridity variables from the Global Aridity and PET Database (http://www.cgiar-csi.org/data/global-aridity-and-pet-database) (*Zomer et al., 2008*). We did not include the altitude layer because a previous study highlighted that viviparous and oviparous species did not exhibit differences in elevational range size (*Lambert & Wiens, 2013*). All layers were clipped to limits based on the distribution of all species in the group combined.

### Ecological niche modeling

Based on the large number of layers included, we performed a preliminary analysis with MaxEnt v.3.4.1 (*Phillips, Anderson & Schapire, 2006*; *Phillips & Dudik, 2008*) for all species using all layers and with default settings featuring cloglog output. Using the jackknife test implemented in MaxEnt, we chose only those variables with high relative importance (11 for each species). In order to avoid collinearity and model overfitting, we extracted the environmental information for each grid cell from this reduced set of layers to perform a Pearson correlation. We then retained only layers with low correlation ($r < 0.75$), and we chose (wherever possible) layers that measured extreme conditions in the case of highly correlated variables—these condition the range limits of species (*Sexton et al., 2009*) as well as the most biologically meaningful layers for this group of species. This species group has a fall-winter reproduction cycle, and relationships between local extinctions and the increase in temperatures by global warming in the reproductive season have been noted (*Sinervo et al., 2010*). The most evident layers with biological meaning for this species group were related to fall and winter seasons, which are the driest and coldest seasons throughout nearly the entire distribution range of studied species. Finally, we chose 11 layers: Max Temperature of Warmest Month (Bio5), Mean Diurnal Range (Bio2), Mean Temperature of Wettest Quarter (Bio8), Mean Temperature of Driest Quarter (Bio9), Precipitation Seasonality (Bio15), Precipitation of Warmest Quarter (Bio18), Precipitation of Coldest Quarter (Bio19), Average Potential Evapotranspiration in May (PET5), Average Precipitation in May (Prec5), Average Precipitation in October (Prec10), and Average Maximum Temperature in January

(Tmax1). The clip of layers, the extraction of climatic information, and Pearson correlations were performed using R (*R Development Core Team, 2017*) and the Raster library (*Hijmans, 2017*).

Final MaxEnt analysis were performed for each species using the default settings with cloglog output and 10 replicate runs employing different random seeds with 80% of the localities for model training and 20% for model testing and bootstrap as replicated run type. As the default threshold-independent receiver operating characteristic (ROC) analyses (*Phillips, Anderson & Schapire, 2006*) of MaxEnt must only be used with true absences and not pseudo-absences or background points (*Lobo, Jiménez-Valverde & Real, 2007*), we avoided this option. We performed statistical evaluation utilizing partial ROC analyses (*Peterson, Papes & Soberón, 2008*) that account for a user-defined maximum acceptable error of omission. Furthermore, we performed partial ROC analyses with *Tool for Partial-ROC* (*Narayani, 2008*) using 50% of the evaluation points resampled in 1,000 bootstrap runs with a fixed error of commission ≤5% (*1-omission threshold >0.95*). Then, a Z-test was conducted to determine whether partial AUC proportions were better than random (AUC = 1.0).

## Phylogeny of the *Sceloporus torquatus* group

*Leaché et al. (2016)* estimated a phylogenomic tree of the *Sceloporus* genus confirming the monophyly of the *torquatus* group in relation to the *megalepidurus* group, which resolved certain taxonomic inconsistencies owing to fewer loci used in previous studies, and for the rapid radiations observed for certain groups of species, what had caused the two main clades in the *torquatus* group to be considered two different groups of species: (1) *torquatus* group including *S. torquatus* and its sister species; and (2) the now former *poinsetii* group that sustains *S. poinsetti* and its sister species (*Leaché, 2010*; *Wiens et al., 2010a*). Unfortunately, *Leaché et al. (2016)* only included 15 species, and likely misidentified two of them. The specimen, UTAR 39870, referred to *S. serrifer* from south Texas, which was recuperated as the sister species of *S. cyanogenys* in the phylogenomic tree of *Leaché et al. (2016)*, corresponding to *S. cyanogenys* according to *Martínez-Méndez & Méndez De La Cruz (2007)*; therefore, it should have no affinity with *S. serrifer* populations from Guatemala and the Yucatan peninsula in Mexico. Likewise, specimen UWBM 6636, identified as *S. mucronatus,* is likely *S. omiltemanus* as the organism was collected approximately 10 km east of the type locality (*Smith, 1939*) in the Sierra Madre del Sur. Furthermore, evidence suggests that this species does not have a close phylogenetic relationship with *S. mucronatus* (*Martínez-Méndez & Méndez De La Cruz, 2007*), which occurs in the Trans-Mexican Volcanic Belt.

In order to estimate the phylogeny of the *S. torquatus* group and include the maximum number of species, we used sequences from four mitochondrial genes (12S, 16S, Nd4, and ND1), and four nuclear genes (RAG1, BDNF, R35, and PNN) retrieved from GenBank (Table S1) for the 23 species recognized for the group, including a new species (MX14-4) from Central West Mexico and three species of the *grammicus* group as an outgroup (*S. grammicus*, *S. heterolepis*, and *S. palaciosi*). As previously highlighted, we made use of the *grammicus* group—the second outgroup of *torquatus*—owing to problems of
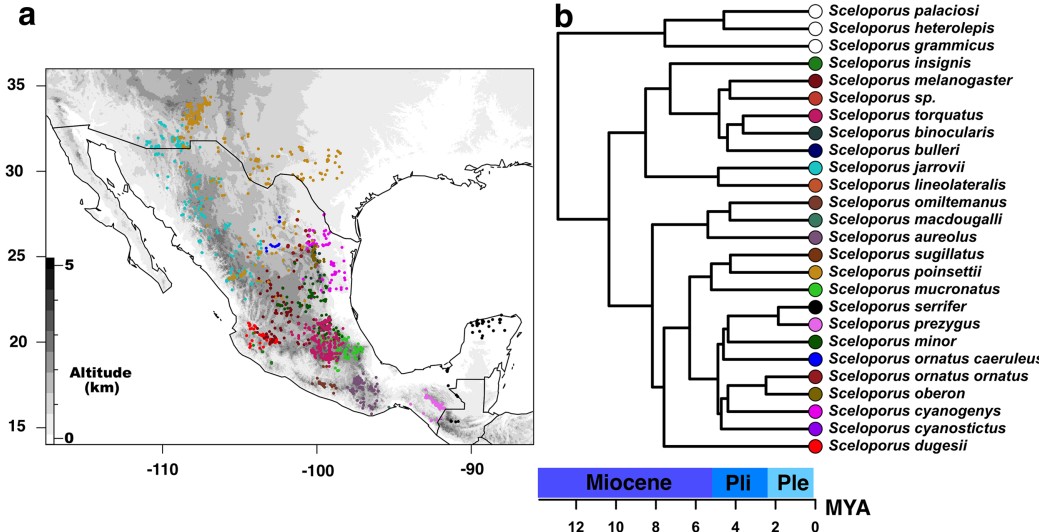

**Figure 1** **Current distribution of species of the *Sceloporus torquatus* group.** (A) Current distribution of species of the *Sceloporus torquatus* group. Darker colors indicate higher elevations, and colored dots in the map show the localities for each species before the final debugging (to get localities in distinct grids and without climatic outliers); also, each color corresponds with the same species in the calibrated tree. (B) Ultrametric time calibrated tree of *S. torquatus* group.

monophyly in *torquatus* with respect to its sister group, *megalepidurus* (*Leaché, 2010*; *Wiens et al., 2010a*; *Leaché et al., 2016*).

Alignment of each locus was performed using Clustal X ver. 2.1 (*Larkin et al., 2007*), and loci were concatenated and refined by eye in Mesquite ver. 3.2 (*Maddison & Maddison, 2017*). A total of 21 partitioning schemes were considered: by the gene region of 12S, 16S, and Nd4-tRNAs, and by the codon position of the remaining nuclear and mitochondrial loci. In order to determine the best substitution model for each data partition, we used jModeltest ver. 2 (*Darriba et al., 2012*) based on the corrected Akaike information criterion (AIC). The models with a parameter for invariant sites (I) in addition to among site-heterogeneity ($\Gamma$) were not considered based on the correlation of these two parameters not allowing for independent optimization (*Sullivan, Swofford & Naylor, 1999*; *Rannala, 2002*). Moreover, the phylogenetic relationships of *torquatus* members group were assessed using maximum likelihood (ML) and Bayesian inference (BI). ML analysis was performed in RAxML ver. 8.1. (*Stamatakis, 2014*) using GTA+ $\Gamma$, and base frequencies that were estimated and optimized for the partitioning scheme listed above with 1,000 non-parametric bootstrap replicates employing the rapid-bootstrapping algorithm. BI was performed with MrBayes ver. 3.2.6 (*Ronquist et al., 2012*) featuring partitioned data using models suggested by jModeltest. However, when the model was not implemented in MrBayes, we used the nearest and most inclusive parameter-rich model for analyses. Four metropolis-coupled MCMC chains were run for 10 million generations, with trees sampled every 1,000 iterations using default temperatures for chain heating. Following a burn-in of 25%, as determined by visualizing posterior distributions of parameter values in Tracer ver. 1.6 (*Rambaut et al., 2014*), we generated a 50% majority rule consensus tree with SumTrees ver. 3.3.1, which is part of

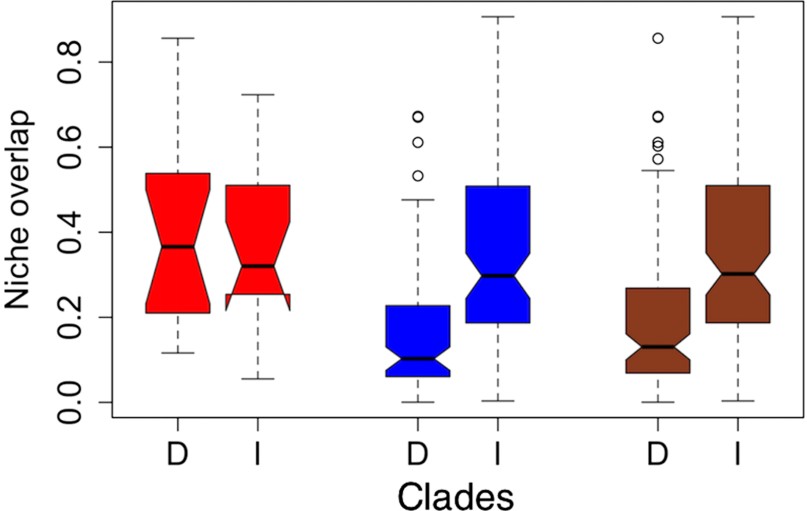

**Figure 2** Notched boxplots for niche overlap indices in terms of Schoener's *D* (*D*) and Warren's *I* (*I*) for the former clades *torquatus* (red) and *poinsettii* (blue), and for total tree (brown). The indices vary between zero (no overlap) to one (complete overlap). Boxes delimit interquartile ranges (25th and 75th percentiles) around the median, whiskers delimit ≈ 2 standard deviations, dotted line indicated maximum and minimum values, and the outliers are represented with circles. Each notch represents the confidence interval of 95% for the median, and lack of overlap between notches is evidence of significant differences between medians.

the DendroPy Python library (*Sukumaran & Holder, 2010*). The plot of the majority rule consensus tree with posterior probabilities and bootstrap proportions from ML analysis was assessed using the ape package (*Paradis, Claude & Strimmer, 2004*) in R (*R Development Core Team, 2017*) (Fig. 1).

In order to obtain a calibrated ultrametric tree for subsequent phyloclimatic analyses, we used the ape R package (*Paradis, Claude & Strimmer, 2004*) to edit the majority rule consensus tree. First, those species excluded from niche analysis were pruned using the *drop.tip* function; then, the tree was made ultrametric and node ages were estimated using a semi-parametric method based on penalized likelihood via the *chronos* function with default settings (Fig. 2). We used divergence values from the phylogenomic analyses of *Leaché et al. (2016)* and *Martínez-Méndez, Mejía & Méndez-De La Cruz (2015)* as calibration points between *torquatus* and *poinsetti* clades (8.24–12.65 MYA) and between *S. serrifer* and *S. prezygus* (1.58–6.35 MYA), respectively.

## Phylogenetic signal of climatic variables and testing for phylogenetic niche conservatism

Following *Münkemüller et al. (2015)*, we assumed an over-simplification of the reality that species niches can be described by single continuous traits (bioclimatic variables). Despite criticism of the methods we adopted, two practical positions were taken to investigate the presence of PNC: (1) PS is used to measure PNC only if the analyzed character evolves under a Brownian motion (BM) mode of trait evolution (*Felsenstein, 1985*); and (2) if, under the exploration of alternative evolutionary models, such as the Ornstein–Uhlenbeck (OU) model (*Butler & King, 2004*), we obtain support for either a

single optimum with high selection strength or a multi-optima OU model with relatively few peak shifts.

To achieve this, first we calculated environmental means for the chosen bioclimatic variables for each species using the phyloclim package (*Heibl & Calenge, 2015*), and then we tested for PS using the package, phytools (*Revell, 2012*), by calculating Blomberg's K (K) (*Blomberg, Garland & Ives, 2003*) with 1,000 simulations and Pagel's lambda (λ) (*Pagel, 1999*) via the ML method. K is a scaled ratio of the variance of data between species and the mean squared error based on the variance–covariance matrix of the phylogeny under a BM expectation with values ranging from zero to infinity, where $K > 1$ indicates a strong PS with the variance distributed between clades and $K < 1$ indicates weak PS with variance within clades (*Blomberg, Garland & Ives, 2003*; *Münkemüller et al., 2012*). Pagel's λ is a scaling parameter for the phylogeny that measures the correlation of observed trait data between species under a BM, whose values range from 0 (no correlation) to 1 (correlation between species), suggesting that phylogenetic relationships suitably predict the pattern of trait evolution (BM process), and that different degrees of PS are included in $0 < λ < 1$ values (*Pagel, 1999*; *Münkemüller et al., 2012*). Moreover, we used the geiger R package (*Harmon et al., 2008*) to test the fit of the data to four alternative models of trait evolution: (1) BM, in which a trait along the phylogeny of a group evolves in a random walk with a constant increase of variance and an expected mean equal to zero (*Felsenstein, 1985*); (2) OU, where traits evolve to an adaptive optimum or around one or some optimal values (*Butler & King, 2004*). However, this should not be interpreted as stabilizing selection in comparative studies (*Cooper et al., 2016*); (3) Early burst or rapid trait evolution followed by stasis (*Harmon et al., 2010*); and (4) Pagel's delta (δ) (*Pagel, 1999*), which models changes to evolution rates through time, where $δ < 1$ is indicative of a slowdown in the recent evolution of a group with trait evolution being concentrated at the base of the phylogenetic tree, and $δ > 1$ is indicative of recent evolution being rapid, and trait evolution being concentrated at the tips of the tree. The identification of best-fitting model of trait evolution was assessed using log likelihood values and AICc, where the model with the higher log likelihood and lower AICc is deemed the better fit (*Hurvich & Tsai, 1989*). Additionally, to clarify our ability to distinguish among models, we followed *Burnham & Anderson (2002*, *2004)*, who pointed out that models with ΔAIC <2 (AIC differences) are more or less equivalent, while models with ΔAIC within 4–7 are distinguishable, and models with ΔAIC >10 are different. We then compared ΔAIC between the model with the lowest AICc and remaining models and established that: ΔAIC <2 = *e* (equivalent models); ΔAIC ≥2 and <7 = *(more or less distinguishable models); ΔAIC ≥7 and <10 = **(distinguishable models); and ΔAIC ≥10 = ***(different models). Following the recommendations of *Münkemüller et al. (2015)*, the white noise model equivalent to no PS was not considered as it exhibited the same pattern as an OU model with strong attraction strength (tends to infinity). Additionally, we performed a multivariate analysis of the bioclimatic variables, tested for PS, and calculated the best trait evolution model for each of the retained axes. An explanation of the methods and general discussion of this analysis can be found in the Supplementary Analysis section.

We also performed a test under a multiple-optima OU framework to infer location, magnitude, and the number of possible adaptive shifts with the bayou R package (*Uyeda & Harmon, 2014*), which utilizes a reversible-jump Bayesian method to test for multiple optima. We first established a prior function with a half-Cauchy distribution prior to $\alpha$ and $\sigma^2$, a normal prior to $\theta$, a conditional Poisson for the number of shifts, and a maximum of one shift per branch. In total, we ran two chains for $2 \times 10^6$ generations, with sampling at every 200 steps. After discarding the first 50% of generations as burn-in, convergence was assessed using Gelman and Rubin's $R$ statistic ($R \leq 1.1$).

To explore the presence of PS in niche overlap patterns (niche evolution), we used the modification of *Warren, Glor & Turelli (2008)* for the age-range correlation (ARC) proposed by *Fitzpatrick & Turelli (2006)*. This method employs a linear regression of node age given the niche overlap of the species, where a positive or negative significant correlation is an indication of PS in niche evolution, and can also be indicative of speciation mode. For this purpose, we used MaxEnt outputs and calculated niche overlap by means of Schoener's *D* and Warren's *I* (a modification of Hellinger distance *I*) statistics, both of which range from 0 (no overlap) to 1 (total overlap) (*Warren, Glor & Turelli, 2008*). The differences between these two metrics are that *I* tended to yield higher values than Schoener's *D*, though the latter assumes that the probability assigned by the ecological niche model to any cell is proportional to species density, which is likely incorrect (*Warren, Glor & Turelli, 2008*). Finally, Warren's *I* statistic was chosen for ARC, and 1,000 iterations of Monte Carlo resampling of the overlap matrix was used to determine the significance of analyses. Niche overlap statistics and ARC analyses were performed with the package phyloclim (*Heibl & Calenge, 2015*) for R.

## Predicted niche occupancy and ancestral tolerances

In order to reconstruct the evolutionary history of niche tolerance or predicted niche occupancy (PNO), we used the methodology of *Evans et al. (2009)*. This method relates suitability scores within the distribution (from MaxEnt analyses) of each species for each bioclimatic variable in order to obtain a unit area histogram of suitability, which represents the tolerance (occupancy) of the species at a given bioclimatic variable (PNOs profiles). Later, PNOs and the pruned phylogenetic tree were made use of to estimate the ancestral tolerance of nodes to each bioclimatic variable using 1,000 random iterations from PNO profiles and a ML method. Additionally, we employed the weighted means of the PNOs in a phylogenetic principal components analysis (pPCA; *Revell, 2009*) to explore a possible climatic differentiation or geographic association between species and clades; however, this method assumes that all traits evolved under a multivariate BM process (*Revell, 2009*; *Uyeda, Caetano & Pennell, 2015*). PNO profiles and ancestral tolerances were calculated using the phyloclim package (*Heibl & Calenge, 2015*), and pPCA was performed using the phytools package (*Revell, 2012*) for R.

Finally, we used an analysis of relative DTT (*Harmon et al., 2003*) to assess the time pattern of niche evolution and how niche disparity is distributed among or within subclades. Here, disparity is the average of the squared Euclidian distance of weighted mean PNO values among all pairs of species (pairwise differences), while relative disparity

is the disparity within a clade divided by the disparity of the entire phylogenetic tree. DTT is calculated as the mean relative disparity of all clades with ancestral lineages present at each speciation event. Then, a null or expected DTT distribution is created using simulated data under a BM model of trait evolution. The expected DTT and observed DTT values of each subclade were then plotted against divergence times to obtain a DTT plot. The results of DTT analyses were subsequently quantified using morphological disparity index (MDI), which is the difference between the observed and expected DTT. Positive MDI values indicate either trait disparity being distributed within subclades or the recent evolution of a trait with divergence between subclades. Conversely, negatives values indicate either a disparity distributed between subclades and the early evolution of a trait or conservatism within deeper clades (*Evans et al., 2009*). We present MDIs for the entire phylogeny and for the former *poinsettii* and *torquatus* clades. The DTT analyses were carried out with the geiger package (*Harmon et al., 2008*) for R with 1,000 simulations and a confidence level of 95%.

## RESULTS

### Ecological niche modeling

Presence data for *Sceloporus sp.* (MX14-4), *S. lineolateralis*, and *S. macdougalli* were excluded from the niche analyses as these species possessed a reduced number for useful points following depuration (<5). For all remaining species, mean AUC scores were >0.75, and were statistically significant with the AUC proportions of partial ROC analyses >1; then, the ecological niche models (Fig. S1) were considered suitable for use as inputs in subsequent analyses.

### Phylogeny of the *Sceloporus torquatus* group

Our reconstructed phylogeny of the *torquatus* group (Fig. S2) is similar to that of previous studies (*Wiens & Reeder, 1997*; *Martínez-Méndez & Méndez De La Cruz, 2007*; *Leaché et al., 2016*), with two main clades corresponding to the former *poinsettii* and *torquatus* groups *Leaché, 2010*; *Wiens et al., 2010a*). Here, we refer to these two clades as *poinsettii* and *torquatus* clades to avoid confusion with the total *torquatus* group, both of which have strong support (*poinsettii* clade: PP = 1, BSP = 100%; *torquatus* clade: PP = 0.99, BSP = 99%). However, as mentioned previously, various differences exist between our phylogeny and that of *Leaché et al. (2016)*: (1) the probable misidentification of *S. omiltemanus* as *S. mucronatus*, where *Wiens & Reeder (1997)* and *Martínez-Méndez & Méndez De La Cruz (2007)* reported the non-monophyly of *S. mucronatus* subspecies, with the latter proposing that *S. mucronatus omiltemanus* be elevated to full species status; (2) the consideration of UTAR 39870 from Texas as *S. serrifer* as, according to *Martínez-Méndez & Méndez De La Cruz (2007)*, populations from Texas and the Northeast of Mexico were considered to be *S. serrifer plioporus* for *Olson (1987)*, being synonymized into *S. cyanogenys*; (3) we included the new specimen MX14-4 (*Sceloporus* sp.), which was resolved as a sister species of *S. melanogaster* with strong support from Bayesian analyses (PP = 1, BSP <75%).

**Table 1 Results of tests for phylogenetic signal of bioclimatic variables used in the study by means of Blomberg's K (K) and Pagel's lambda (λ) values.**

| Bioclimatic layer | Blomberg's K | | Pagel's lambda (λ) | | | |
|---|---|---|---|---|---|---|
| | K | p | λ | logL | logL0 | p |
| Mean diurnal range (Bio2) | 0.752 | 0.077 | 0.782 | −99.895 | −100.465 | 0.285 |
| Max temperature of warmest month (Bio5) | 0.609 | 0.332 | 8.06E-05 | −111.561 | −111.561 | 1 |
| Mean temperature of wettest quarter (Bio8) | 0.583 | 0.454 | 8.06E-05 | −114.106 | −114.105 | 1 |
| Mean temperature of driest quarter (Bio9) | 0.606 | 0.365 | 6.61E-05 | −113.723 | −113.723 | 1 |
| Precipitation seasonality (Bio15) | **0.979** | **0.003** | **0.899** | **−92.018** | **−95.390** | **0.009** |
| Precipitation of warmest quarter (Bio18) | 0.627 | 0.307 | 8.06E-05 | −138.547 | −138.547 | 1 |
| Precipitation of coldest quarter (Bio19) | 0.530 | 0.683 | 8.06E-05 | −106.060 | −106.060 | 1 |
| Average potential evapo-transpiration in May (PET5) | 0.936 | 0.245 | 8.06E-05 | −96.862 | −96.862 | 1 |
| Average precipitation in May (Prec5) | 0.902 | 0.127 | 0.154 | −109.906 | −110.043 | 0.6 |
| Average precipitation in October (Prec10) | 0.896 | 0.168 | 8.06E-05 | −113.397 | −113.396 | 1 |
| Average maximum temperature in January (Tmax1) | 0.779 | 0.515 | 0.722 | −117.589 | −117.482 | 1 |

**Note:**
Traits with values significantly different from zero are in bold.

## Phylogenetic signal of climatic variables and testing for phylogenetic niche conservatism

In the present study, tests of PS indicated that only precipitation seasonality (Bio15) exhibited significant support for the PNC hypothesis (Table 1) with a moderate to weak PS and variance distributed within clades ($K = 0.9789271$, $p = 0.003$), thereby suggesting a high correlation of the data with a BM process ($\lambda = 0.8990152$, $p = 0.009$). This coincides with the test of alternative models of evolution (Table 2), where Bio15 shows support for BM evolution based on the difference between alternative models being just over two ($\Delta AIC = 2.0003$). The other bioclimatic layer showing BM evolution with a lower AICc was indistinguishable from other models of evolution as well as equivalent models (i.e., BM and δ are equivalents in Bio2 and Tmax1). Similarly, only Precipitation of Coldest Quarter (Bio19) presented evidence for weak support of an OU model of evolution as the difference in the alternative model was mild ($\Delta AIC = 2.13$); however, selection strength was relatively weak ($\alpha = 0.597$; Table S3). This likely implies weak support for the PNC hypothesis for Bio19 when using the interpretation of *Münkemüller et al. (2015)*, where PNC is indicated by relatively strong selection strength and one or relatively few adaptive peak shifts. The remaining bioclimatic layers with OU showing lower values of AICc are indistinguishable from other models of trait evolution tested, possibly because of the limited sample size. Notably, in all cases, Pagel's delta (δ) was >1 (Table 2), indicating a tendency for trait evolution to be concentrated in the tips of the tree. Likewise, the multi-optima OU method implemented in Bayou failed to correctly detect the location and magnitude of adaptive shifts (Table S4; Fig. S3) as the mean number of shifts was nine ($K = 9$), and parameters are correctly estimated only if the number of shifts is not large ($K > 25\%$ the number of tips) (*Uyeda & Harmon, 2014*).

On average, niche overlap values (Fig. 2) were low (Schoener's *D* and Warren's *I* statistics <0.4) for all species and within *torquatus* and *poinsettii* clades. Similarly, just a

**Table 2 Performance of alternative evolution models for each bioclimatic variable.**

| Bioclimatic layer | Model | lnL | AICc | Parameters |
|---|---|---|---|---|
| Mean diurnal range (Bio2) | BM | −100.179 | 204.959 | 2 |
| | δ = 2.36 | −99.578 | 206.420 | 3 e |
| | OU | −99.696 | 206.656 | 3 e |
| | EB | −100.179 | 207.622 | 3* |
| Max temperature of warmest month (Bio5) | OU | −111.561 | 230.385 | 3 |
| | δ = 2.89 | −112.214 | 231.692 | 3 e |
| | BM | −114.401 | 233.402 | 2* |
| | EB | −114.401 | 236.065 | 3* |
| Mean temperature of wettest quarter (Bio8) | OU | −114.106 | 235.474 | 3 |
| | δ = 3.00 | −114.890 | 237.042 | 3 e |
| | BM | −117.431 | 239.461 | 2* |
| | EB | −117.431 | 242.124 | 3* |
| Mean temperature of driest quarter (Bio9) | OU | −113.603 | 234.470 | 3 |
| | δ = 2.99 | −113.763 | 234.790 | 3 e |
| | BM | −115.625 | 235.850 | 2 e |
| | EB | −115.625 | 238.513 | 3* |
| Precipitation seasonality (Bio15) | BM | −92.121 | 188.841 | 2 |
| | δ = 1.68 | −91.789 | 190.841 | 3* |
| | OU | −91.975 | 191.213 | 3* |
| | EB | −92.121 | 191.504 | 3* |
| Precipitation of warmest quarter (Bio18) | OU | −138.302 | 283.866 | 3 |
| | δ = 2.89 | −138.406 | 284.076 | 3 e |
| | BM | −139.976 | 284.552 | 2 e |
| | EB | −139.976 | 287.215 | 3* |
| Precipitation of coldest quarter (Bio19) | OU | −106.000 | 219.263 | 3 |
| | δ = 2.91 | −107.070 | 221.402 | 3* |
| | BM | −109.892 | 224.385 | 2* |
| | EB | −109.892 | 227.048 | 3** |
| Average potential evapo-transpiration in May (PET5) | BM | −97.561 | 199.722 | 2 |
| | δ = 2.96 | −96.752 | 200.768 | 3 e |
| | OU | −96.824 | 200.912 | 3 e |
| | EB | −97.561 | 202.385 | 3* |
| Average precipitation in May (Prec5) | OU | −109.726 | 226.716 | 3 |
| | δ = 2.97 | −109.975 | 227.214 | 3 e |
| | BM | −111.658 | 227.917 | 2 e |
| | EB | −111.658 | 230.580 | 3* |
| Average precipitation in October (Prec10) | OU | −112.753 | 232.769 | 3 |
| | δ = 2.89 | −112.943 | 233.149 | 3 e |
| | BM | −114.923 | 234.669 | 2 e |
| | EB | −114.923 | 237.109 | 3* |
| Table 2 (continued). | | | | |
|---|---|---|---|---|
| **Bioclimatic layer** | **Model** | **lnL** | **AICc** | **Parameters** |
| Average maximum temperature in January (Tmax1) | BM | −117.713 | 240.026 | 2 |
| | δ = 2.78 | −116.846 | 240.956 | 3 *e* |
| | OU | −117.057 | 241.378 | 3 *e* |
| | EB | −117.713 | 242.689 | 3* |

**Notes:**
The differences between the model with lower AICc and the rest of the models are indicated with fallow abbreviations: *e*, equivalent models.
* More or less distinguishable models.
** Distinguishable models.

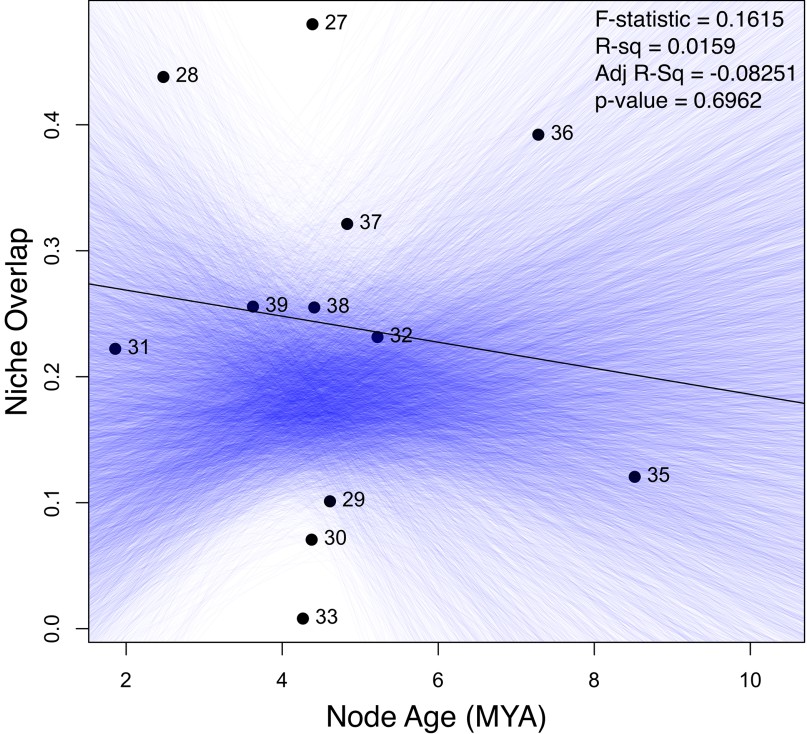

**Figure 3 Linear regression of the age-range correlation (ARC).** Abscissa axis corresponds with node age and ordinate axis with Warren's *I* niche overlap index. Blue lines correspond with regression lines from Monte Carlo randomization. 

few pairs of species exhibited moderate to high values (Table S5), including *S. cyanostitctus* vs *S. ornatus caeruleus* (Warren's *I* = 0.907). However, none of these included sister taxa or closely-related species in our analyses, with exception of the small clade formed by *S. cyanogenys* + (*S. oberon* + *S. ornatus ornatus*), which exhibited Warren's *I* statistic values from 0.753 to 0.894. Moreover, the ARC showed no significant correlation between niche overlap at internal nodes and divergence time (Fig. 3) and failure to detect PS in niche evolution in all bioclimatic layers used, which is consistent with a lack of PS in nearly all individually tested bioclimatic layers except Bio15.

## Predicted niche occupancy and ancestral tolerances

Predicted niche occupancy profiles (Fig. 4) exhibited high heterogeneity in certain bioclimatic variables (e.g., Tmax1, Bio2, Bio5, and Bio9), with species occupying different sections of parameter space and with varying levels of specificity in climatic tolerance—as denoted by the breadths of the profiles. However, various overlapping peaks indicate similar climatic tolerance between a few species across all bioclimatic layers, which were especially important in Average Potential Evapotranspiration in May (Pet5) and in Precipitation of the Coldest Quarter (Bio19). Additionally, Bio19 exhibited the narrowest overall PNO profile breadth of all bioclimatic layers, which is apparently consistent with an OU model of trait evolution with a single optima as detected for this bioclimatic layer (Table 2), though the AIC$_c$ difference between OU and Pagel's delta ($\delta$) is merely 2.12. It is also important to note the case of *S. serrifer,* which exhibits the most extreme values for Mean Temperature of Wettest Quarter (Bio8) and Mean Temperature of Driest Quarter (Bio9) PNO profiles. Plots detailing the history of climatic tolerance evolution (Fig. 5) exhibit no pattern between the two main clades, with crossing branches from different clades for all bioclimatic variables—indicating divergent evolution—and only some nearly overlapping nodes were recovered, which is indicative of a grade of convergent climatic origins. However, these plots were built under the assumption of BM evolution; therefore, only the plot for Bio15 would have a non-biased interpretation. Though given the narrow differences in AIC$_c$ values among BM and the remaining models of trait evolution, we cannot rule out BM as the trait evolution mode in all analyzed layers except Bio19, which is supported by DTT analyses (explained in the following section). Nevertheless, the means are close and density of climate tolerance is more or less narrow for each species on Prec10, Bio9, Bio18, and Bio19. However, in the Bio19 plot, only the branch of *S. serrifer* and *S. prezygus* demonstrates major divergent evolution.

Phylogenetic principal components analysis exhibited no pattern or separation between clades (Fig. S4), with certain species being more influenced in their distribution by Bio2 and Pet5 (*S. cyanogenys*, *S. ornatus*, *S. poinsetti*, and *S. jarrovii*), while others were more strongly influenced by Bio15 and Prec5 (*S. aureolus*, *S. mucronatus*). Again, *S. serrifer* exhibited the more divergent niche, which was primarily influenced by Bio9 and Bio19. As the pPCA analysis did not present an evident pattern or separation between clades, phylogenetic MANOVA analysis was not necessary to confirm any significant differences. Nevertheless, this method is useful for visualizing divergence across a phylomorphospace, and the interpreted contribution of each trait must be considered with caution owing to the assumption of BM evolution for all traits (*Uyeda, Caetano & Pennell, 2015*).

Results from the analysis of relative DTT indicated (Fig. 6) that nearly all bioclimatic layers possess zero disparity in internal (deep) nodes, which is indicative of early conservatism in major clades. Moreover, the majority of bioclimatic layers were not outside of the 95% CI of the null BM model (the space between dotted line and gray shaded area in Fig. 6), though some peaks indicated a tendency for slight divergence in recent nodes, as well as mild evolution within clades that was consistent with that of positive

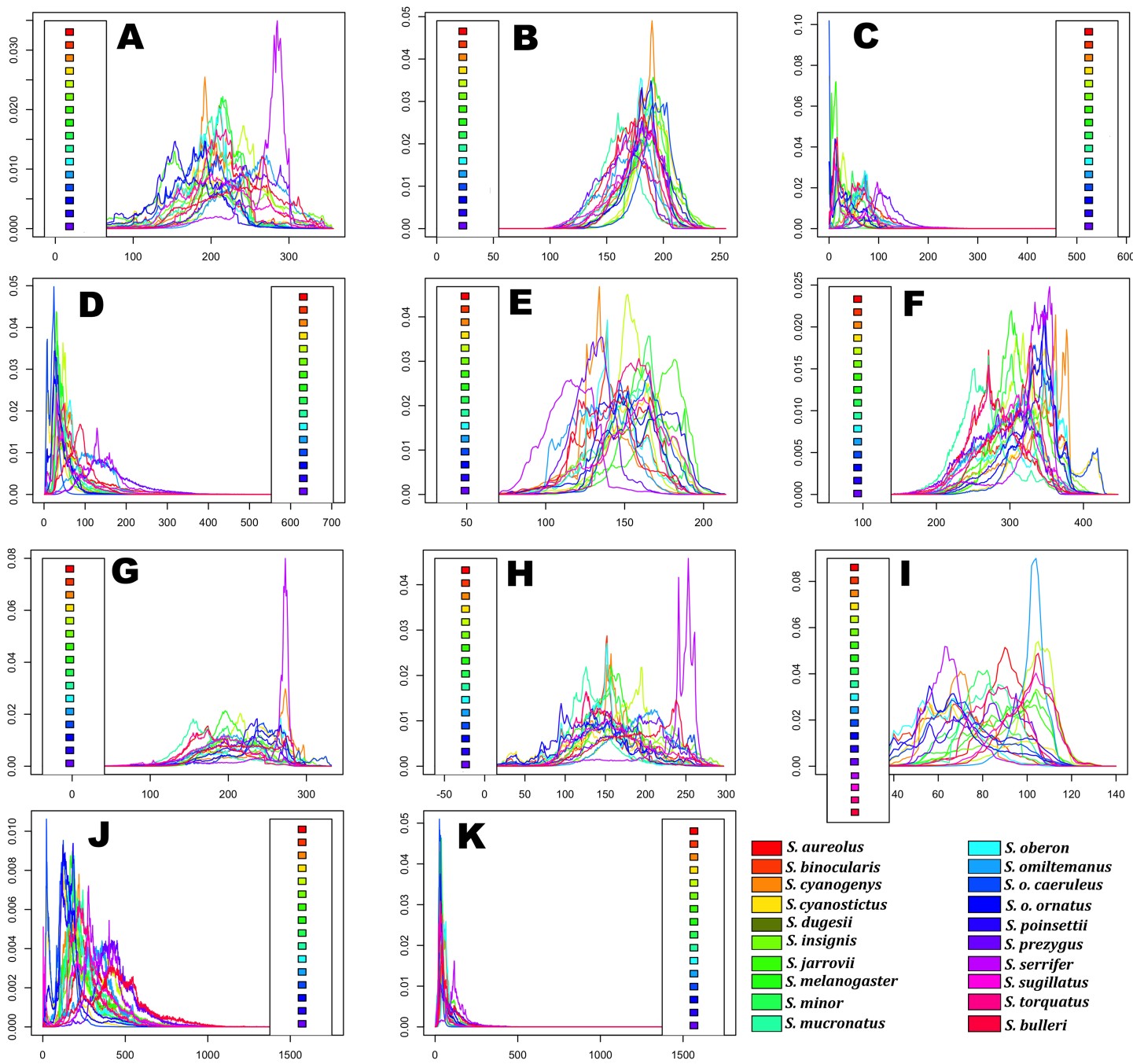

**Figure 4 Predicted niche occupancy (PNO) profiles for *Sceloporus torquatus* species group.** Horizontal axes represent the bioclimatic variable parameter and vertical axes indicate the total suitability of the bioclimatic variable index for each species over its geographic distribution. Overlapping peaks indicate similar climatic tolerances, and the breadth of the profile indicates the climatic tolerance specificity. Species names consisting of the four letters of the species epithets, except for *Sceloporus ornatus caeruleos* (caeru). (A) Tmax1, (B) Pet5, (C) Prec5, (D) Prec10, (E) Bio2, (F) Bio5, (G) Bio8, (H) Bio9, (I) Bio15, (J) Bio18, and (K) Bio19.

MDI values. However, exceptions including Bio5, Bio8, and Bio9 showed levels outside of the 95% CI of null speciation in relatively recent times. In appearance, this result was contradictory for Bio19, which exhibited support for OU evolution (barely distinguishable

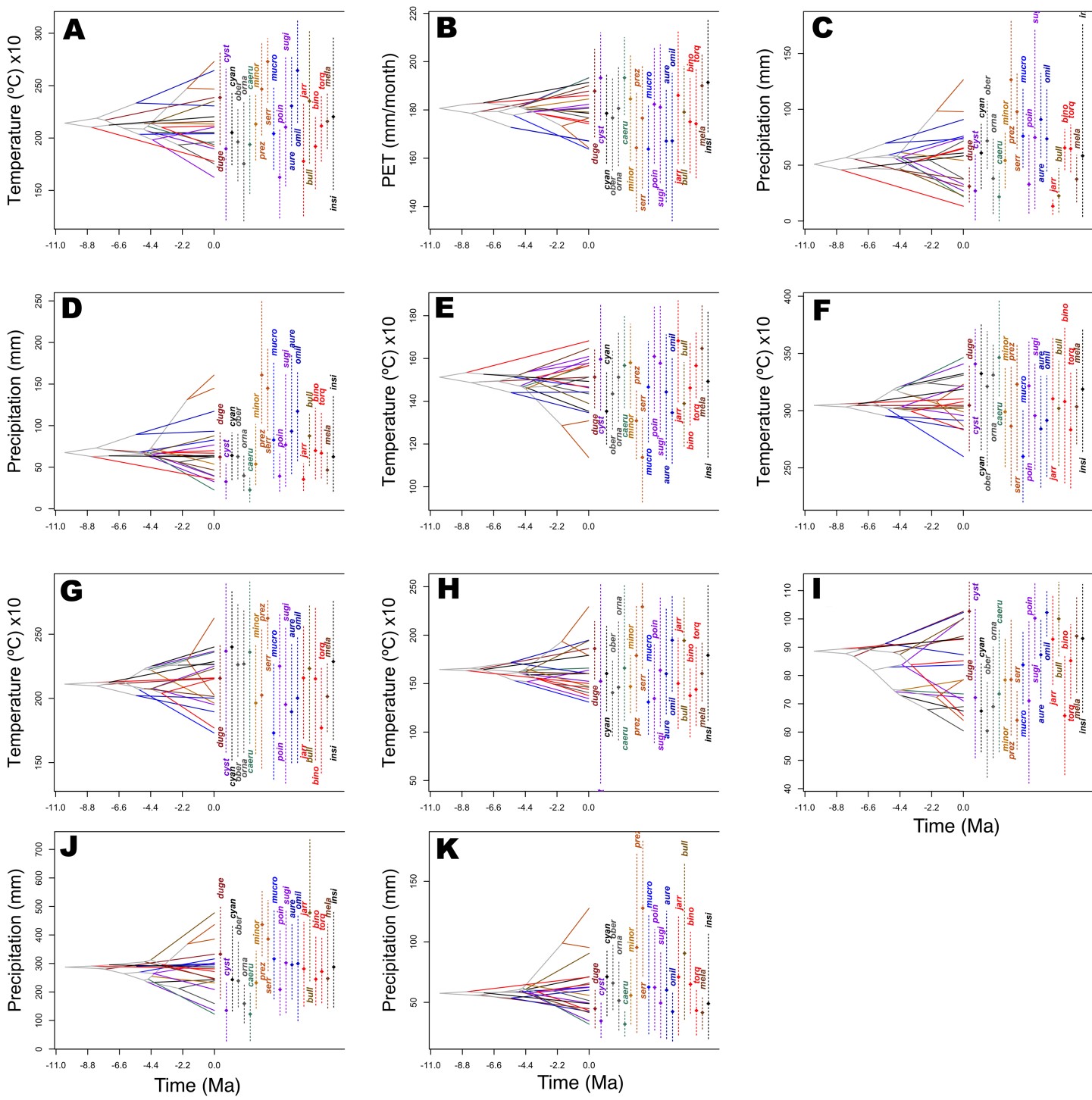

**Figure 5 History of evolution of climatic tolerances for *Sceloporus torquatus* species group.** The chronogram topology of the group is projected into niche parameter space (*y*-axis), and mean climatic tolerances based on 100 random samples of the PNO profiles are represented at internal nodes. Crossing branches of the phylogenetic tree indicate convergent niche evolution among taxa from different clades, and overlapping internal nodes indicate convergent climatic origins. A vertical dashed line indicates the 80% central density of climate tolerance for each species, and the point of the same color indicates the mean. Species names consist of the first three or four letters of the species epithets. (A) Tmax1, (B) Pet5, (C) Prec5, (D) Prec10, (E) Bio2, (F) Bio5, (G) Bio8, (H) Bio9, (I) Bio15, (J) Bio18, and (K) Bio19.     
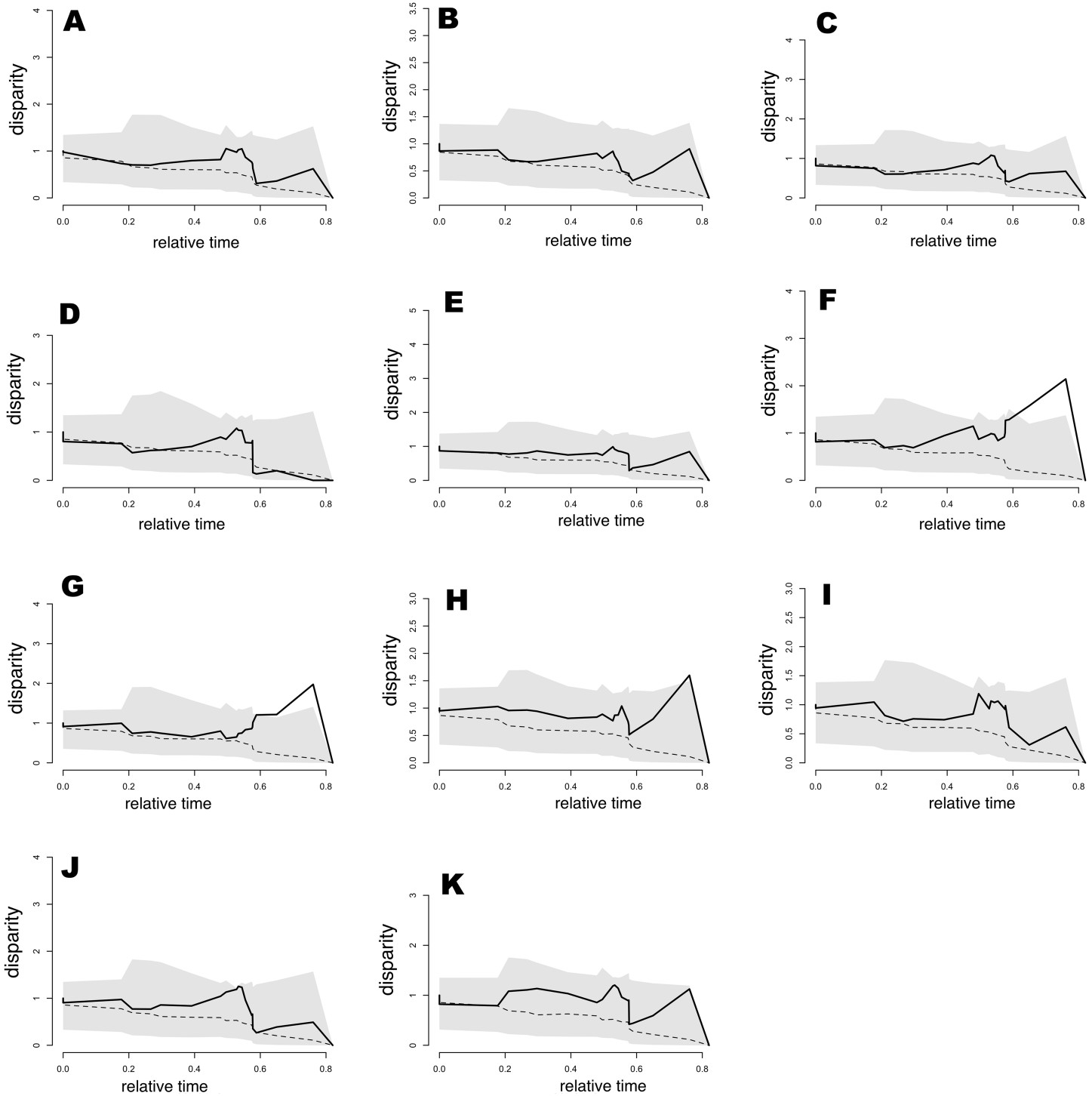

**Figure 6 Plots of accumulation of relative disparity through time (DTT) for climatic tolerances in the *Sceloporus torquatus* species group.** The plot summarizes the distribution of the relative disparity through time (solid line) compared with mean disparity as simulated under 1,000 replicates of an unconstrained model of Brownian Evolution (dashed line). (A) Tmax1, (B) Pet5, (C) Prec5, (D) Prec10, (E) Bio2, (F) Bio5, (G) Bio8, (H) Bio9, (I) Bio15, (J) Bio18, and (K) Bio19.

**Table 3 Morphological disparity index (MDIs) for total phylogeny and for former *poinsettii* and *torquatus* clades.**

| Bioclimatic layer | MDI value | | |
| --- | --- | --- | --- |
| | Total tree | *torquatus* Clade | *poinsettii* Clade |
| Mean diurnal range (Bio2) | 0.177 | 0.062 | 0.165 |
| Max temperature of warmest month (Bio5) | 0.445 | 0.253 | 0.425 |
| Mean temperature of wettest quarter (Bio8) | 0.365 | 0.390 | 0.285 |
| Mean temperature of driest quarter (Bio9) | 0.331 | 0.283 | 0.256 |
| Precipitation seasonality (Bio15) | 0.199 | 0.463 | 0.069 |
| Precipitation of warmest quarter (Bio18) | 0.198 | 0.420 | 0.081 |
| Precipitation of coldest quarter (Bio19) | 0.299 | 0.187 | 0.269 |
| Average potential evapo-transpiration in May (PET5) | 0.151 | 0.032 | 0.090 |
| Average precipitation in May (Prec5) | 0.141 | 0.149 | 0.159 |
| Average precipitation in October (Prec10) | 0.027 | −0.119 | −0.010 |
| Average maximum temperature in January (Tmax1) | 0.157 | −0.080 | 0.140 |

Note:
The morphological disparity index (MDI) value represent the overall difference in disparity between the observed and the unconstrained null hypothesis, MDIs >0 indicate niche evolution and MDIs <0 indicate niche conservatism.

from the alternative model) and was within the 95% CI of null speciation in the DDT plot. However, this was likely an effect of low sample size or relatively weak selection strength. Therefore, as mentioned previously, Bio15 weakly supported BM evolution, though the DDT plot indicated that this variable was always within the 95% CI of BM process null speciation. Generally, the slightly higher levels (but not significant) of disparity in DTT for all bioclimatic layers were concentrated in subclades in relatively recent times (relative time between 0.3 and 0.8).

Morphological disparity index values for the total tree (Table 3) were positive in all cases. However, values were between 0.027 and 0.445, suggesting a low to median level of niche evolution within subclades as well as some niche conservatism between subclades, likely because the majority of low ecological disparity originating in more recent divergence events. This tendency was also consistent with Pagel's delta ($\delta$) values >1 (Table 2). Nevertheless, with the exception of Bio5, Bio8, and Bio9, we cannot refute that the evolution of all bioclimatic variables occurred according to a BM trait evolution process. The same pattern of positive MDI values was generally observed for *torquatus* and *poinsettii* clades, with the exception of low negative MDI values for Prec10 (in both main clades) and Tmax1 (only for the *torquatus* clade), indicating a mild level of early ecological disparity for this bioclimatic variable within those clades.

## DISCUSSION

In their study, *Lambert & Wiens (2013)* provided solid evidence of viviparity among phrynosomatids evolving in colder climates, but their ancillary prediction that viviparous species in tropical montane regions will show signatures of speciation via climatic niche conservatism, with similar climatic niches in allopatric sister, species was not tested. In the present study, we focused on addressing this prediction by means of
a phyloclimatic study using a wide set of climatic variables (as opposed to four bioclimatic layers related to thermal niche in *Lambert & Wiens (2013)*) in order to understand the speciation mechanism in the divergence of the viviparous lizards of the *S. torquatus* group. Our analyses do not fully support the tested prediction as we found low to moderate niche divergence, with no PNC in nearly all bioclimatic variables used for the reconstruction of ecological niche models, and low niche overlap between sister species of the group.

## There are PNC among species of *torquatus* group?

According to the criterion of *Münkemüller et al. (2015)*, among the 11 bioclimatic variables tested, just two (Bio15 and Bio19) and only PC1 scores of the multivariate analysis (see Supplementary Analysis) showed weak evidence of PNC. Generally, we found that the analyses of the individual bioclimatic variables and PC2 scores exhibited a lack of PS and a poor power for distinguishing among evolutionary models with $AIC_C$, in spite of DTT analyses suggesting that we cannot rule out a BM process in the majority of bioclimatic variables.

Some possible explanations exist for the lack of PS observed among some bioclimatic variables, which include (1) evolutionary rate, (2) selection, and (3) sample size:

(1) *Evolutionary rate and PS*. Existing evidence suggest that evolutionary rate does not affect PS for continuous characters when the evolutionary process approximates BM (*Revell, Harmon & Collar, 2008*). Additionally, low PS should not be interpreted as evidence of a high evolutionary rate as PS is merely a measure of a pattern, and its relationship with the evolutionary process is complex (*Revell, Harmon & Collar, 2008*). Nevertheless, a certain level of heterotachy among nearly all bioclimatic variables is possible as observed in DTT plots (despite being within CI limits), which coincides with the high heterotachy detected in Squamata phylogeny and poor statistical fit to BM in the niche evolution of many reptiles (*Pie et al., 2017*). However, PC3 scores did not exhibit PS as this axis demonstrated support for Pagel's delta ($\delta$) with values >1, which could indicate a rapid and recent evolution to dry ambient in the clade that sustains *S. serrifer*. For example, *S. serrifer* populations arrived relatively recently to the north of the Yucatan peninsula (last glacial period), which is primarily comprised of dry forest vegetation (*Martínez-Méndez, Mejía & Méndez-De La Cruz, 2015*). Nonetheless, as we will discuss later, we believe that this result is an effect of the resolution of the bioclimatic layers we used as we have some elements that indicate that the thermal requirements of this species are basically the same between lowland and highland populations, and maybe the requirements regarding humidity could be the same, as well.

(2) *Selection and PS*. Another potential factor for the lack of PS observed in the majority of bioclimatic variables and PC2 scores could be related to some grade of divergent selection as it has been reported that PS is low under this evolutionary process (*Revell, Harmon & Collar, 2008*). Divergent selection can originate because of environmental differences among populations and species, which is in line with
allopatric speciation associated with ecological speciation (*Mayr, 1947*; *Rundle & Nosil, 2005*). A similar scenario was described for zones with high physical and environmental heterogeneity in ecological gradients or tropical montane regions, such as the Mexican mountain ranges where the majority of species in the *S. torquatus* group are distributed. In this scenario, any geographical distance promotes environmental distance between populations, and this ecological distance reduces dispersal and gene flow between adjacent populations, thereby promoting niche divergence and disruptive ecological selection that may produce allopatric speciation wherein the sister species have different niches than the ancestor (*Pyron & Burbrink, 2010*; *Pyron et al., 2015*). Furthermore, the observed lack of PS in the present study coincides with the ecological differences that were detected in PNO profiles (Fig. 4), which exhibited high heterogeneity with low niche overlap values among the majority of sister species in the *S. torquatus* group.

(3) *Sample size and PS*. Low sample size is another likely explanation for the lack of resolution in the test we used to select a trait evolution model and detect PS (excepting for Bio5). Low sample size signifies a problem, specifically for the bayou method as parameters are only correctly estimated if the number of shifts is no larger than the number of tips in the tree ($K > 25\%$ the number of tips) (*Uyeda & Harmon, 2014*), although this could not be a problem with the other tests. Notably, *Cressler, Butler & King (2015)* demonstrated that model selection power can be high between BM and OU models with different selective regimens, even with a small-sized tree, although various model parameter estimations improve with larger sample sizes (i.e., $\alpha$ converge toward the true value as the amount of data increases).

## There are climatic niche similitudes or differences among species of *torquatus* group?

As previously mentioned, low niche overlap values between sister species could represent an additional indicator of no niche conservatism. This contrasts the results of *Warren, Glor & Turelli (2008)*, who observed moderate and high niche overlap and conservatism in many sister species of butterflies, birds, and mammals in Mexico. However, the low niche overlap values observed for the *torquatus* group are not an exception and are expected based on ecological speciation in tropical montane ranges (*Pyron & Burbrink, 2010*; *Pyron et al., 2015*). For example, some studies of freshwater fishes of North America indicated that certain clades presented high niche overlap and conservatism, while others showed high niche diversification and low niche overlap (*McNyset, 2009*; *Culumber & Tobler, 2016*). Similar evidence suggests that sister species of tropical plethodontid salamanders tend to have divergent climatic niches compared to temperate sister species (*Kozak & Wiens, 2007*). Furthermore, a number of studies have highlighted the importance of not only niche overlap in the understanding of diversification, but also the sympatry and range overlap of sister species or closely related species as certain speciation models consider competition for resources to drive sympatric speciation, with ecological differentiation arising to prevent competition (*Rundle & Nosil, 2005*; *Nosil, 2012*).

According to *Losos (2008)*, it is necessary to carefully identify niche similitudes as PNC as—in this case—conservatism originates as a byproduct of a historic process in which no related species share the same geographic range. In this sense, there are lines of evidence that support ecological differentiation in sympatric speciation (*Bush & Smith, 1998*), while other studies underestimate its role—even suggesting that geographic overlap between clades in certain species restricts diversification (*Kozak & Wiens, 2010*). Future studies should focus on whether interactions with other species of lizards could influence the niche evolution of these species.

However, in agreement with the general pattern observed when analyzing individual bioclimatic variables, the absence of a significant correlation between niche overlap at internal nodes and divergence time in the ARC analyses is an additional indication of the absence of PS in many bioclimatic variables during niche evolution of the *torquatus* group. This is also evidence that climatic niche differentiation (ecological divergence) along with PNC in Bio15, Bio19, and PC1 is an important factor in the diversification of the *torquatus* group.

Moreover, PNO profiles show high heterogeneity in climatic occupation levels, which indicates radiation over the spectrum of ecological space represented for the analyzed bioclimatic variables. Nevertheless, some overlapping peaks existed, reflective of similar tolerances in certain species; however, similar tolerances are not shared for the same species in each bioclimatic variable, and no sister species share similar tolerances in all cases except for Bio19, which is linked to the fall-winter reproductive cycle. The most different occupations in PNO profiles were observed in *S. serrifer*, which can be explained by this species occurring in habitats ranging from highlands to nearly sea level. Accordingly, PNO profiles for this species suggest distinct ecological preferences and some degree of ecological differentiation between most of the species without groups of sister species sharing the same ecological niche, as confirmed by PCA and pPCA analyses.

Furthermore, the evolutionary history of climatic tolerances indicated that only certain species had some degree of convergent climatic origins for a number of bioclimatic variables, with most species exhibiting varying magnitudes of divergent evolution. However, Bio19 exhibits the lowest magnitude of final divergence between species of the group, except for the clade formed by *S. serrifer* and *S. prezygus*. In addition, analysis of relative DTT and MDI values indicates that ecological disparity tends to be distributed within subclades rather than between subclades, with some divergence in recent nodes as confirmed by PC3 scores evolution (Pagel's delta ($\delta$) >1). However, with the exception of Bio5, Bio8, and possibly Bio9, we cannot dismiss evolution following a BM process owing to disparities not being outside of the 95% CI of the null BM model. Therefore, evidence of a lack of niche conservatism and some recent accumulation of ecological diversity—especially among particular species (*S. serrifer* clade)—could be associated with possible geographic and climatic isolation throughout speciation, which could promote the rapid accumulation of ecological differences among a few species of the group (*Culumber & Tobler, 2016*). This pattern coincides with the results of *Pie et al. (2017)*, who described an extensive rate of heterogeneity in the climatic niche

evolution of squamates, with shifts involving accelerations concentrated in their recent evolutionary history.

## What drives evolution of climatic niche in *S. torquatus*?

Our results indicated that the niche evolution of the *S. torquatus* group possesses a combination of divergence and PNC, likely linked with disruptive evolution and common physiological requirements that are important for this group of species. The results suggest that climatic variables chosen for estimation of the ecological niche of *torquatus* group had an evident link to the current fall-winter reproductive cycle of viviparous lizards (i.e., Precipitation of Coldest Quarter (Bio19), Average Maximum Temperature in January (Tmax1), and Average Precipitation in October (Prec10)). Likewise, Mean Temperature of Driest Quarter (Bio9) matched the late fall (November) and winter in the Mexican Plateau (Central Mexico) and Chihuahuan Desert zone (*Willmott & Matsuura, 2001*; http://www.worldclim.org), where many species of the *torquatus* group can be found. Moreover, despite a lack of data regarding the reproductive biology and demography of the entire group, the remaining climatic variables could have direct relevance in phases of life history; for example, Average Potential Evapotranspiration in May (PET5), Average Precipitation in May (Prec5), Max Temperature of Warmest Month (Bio5), and Precipitation of Warmest Quarter (Bio18) could be linked with the survival of offspring as parturition for some species in this group has been reported to occur between late April and early May (*Guillette & Méndez-De La Cruz, 1993*; *Méndez-De La Cruz, Villagrán-Santa Cruz & Andrews, 1998*; *Feria-Ortiz, Nieto-Montes De Oca & Salgado-Ugarte, 2001*; *Villagrán-Santa Cruz, Hernández-Gallegos & Méndez-De La Cruz, 2009*), the warmest months in many occurrence sites of the group. This finding coincides with *Watson, Makowsky & Bagley (2014)*, which determined that Max Temperature of Warmest Month (Bio5) is frequently the best predictor of viviparous populations of *Phrynosoma*, *Sceloporus*, and *Plestiodon* in North America. However, although there is an absence of studies on the thermal susceptibility of the young, we assume that they could be more vulnerable than adults to overheating and dehydration because of their small size. This implies that the temperature and humidity range of their activity period should be lower, which would represent a limitation for the establishment of populations in certain areas, though these zones have conditions within the limits of tolerance for adults. Therefore, it would be necessary to conduct studies on the thermoregulation and locomotor performance of young and subadult individuals to determine the role that these stages would have in the establishment of populations. Likewise, Mean Temperature of Wettest Quarter (Bio8) could be related to the ovarian cycle as vitellogenesis in species of this group has been reported to occur throughout the spring and fall (*Guillette & Méndez-De La Cruz, 1993*; *Méndez-De La Cruz, Villagrán-Santa Cruz & Andrews, 1998*; *Feria-Ortiz, Nieto-Montes De Oca & Salgado-Ugarte, 2001*; *Villagrán-Santa Cruz, Hernández-Gallegos & Méndez-De La Cruz, 2009*), which is the wettest period of the year in nearly all distribution areas of the group, linked with the abundance of food necessary for the accumulation of yolk proteins in follicles (*Feria-Ortiz, Nieto-Montes De Oca & Salgado-Ugarte, 2001*). Although the ovarian cycle is highly conserved across different

altitudes in many *Sceloporus* species, the testicular cycle is not and exhibits shifts related to altitude (*Villagrán-Santa Cruz, Hernández-Gallegos & Méndez-De La Cruz, 2009*), possibly linked to the temperature necessary for proper testicular development, accessory sexual structures, and sperm maturation (*Pearson, Tsui & Licht, 1976*; *Van Damme, Bauwens & Verheyen, 1987*; *Villagrán-Santa Cruz, Méndez-De La Cruz & Parra-Gámez, 1994*). Therefore, the variation and plasticity of reproduction cycles must be evaluated—particularly in males—in order to determine the climatic requirements and their importance in the distribution of these species. Likewise, Mean Diurnal Range (Bio2) and Precipitation Seasonality (Bio15) has been reported to have of strong relevance for the evolution of climatic niches in squamate reptiles (*Pie et al., 2017*). This is likely the result of these bioclimatic layers reflecting the extreme conditions of both temperature and humidity, and this coincides with the evidence of PNC in PC1 scores evolution that is weakly linked with temperature variation. The latter is important because previous research has highlighted that extreme climatic conditions could determine the range limits of species (*Sexton et al., 2009*), and maybe this may help understand why we detect PNC in PC1 scores and Bio15. It is probable these species share similar climatic requirements linked with precipitation and, less important, temperature variation, which may be bounded between precise limits, but ecological surveys on these species will be necessary to address this supposition.

On the other hand, the single optimum OU model of evolution for Precipitation of Coldest Quarter (Bio19) could be interpreted as evidence of stabilizing selection (*Hansen, 1997*), although some authors do not recommend the use of this term to refer to evolution around an optimal value (*Cooper et al., 2016*). As such, this optimum is more likely explained by the fact that species aside from those of the *S. torquatus* group are found at sites with different levels of annual precipitation, and precipitation is concentrated in the same season with the driest days being concentrated in the last and first months of the year (*Willmott & Matsuura, 2001*; http://www.worldclim.org); this is probably the reason for PNC detection in this bioclimatic variable. Of note, we must be careful affirming that a single optimum OU process is the best model for Bio19 as the multiple-optima OU analyses fail because of sample size. We believe that the narrow overall breadth of the PNO profile for Bio19, which is indicative of similar levels of tolerance for all species of the group, is indirect evidence of a single optimum OU process. This is the only bioclimatic variable that is directly linked to the fall-winter reproductive cycle that seems to be conserved, which nearly all species of this group similarly require.

Surprisingly, the bioclimatic layers linked with temperature did not show PNC, a relevant issue as temperature during breeding season is the principal factor for estimating extinction probabilities owing to global warming in lizards (*Sinervo et al., 2010*). However, it remains possible that a great number of species in this group have not been thoroughly explored throughout the entire climatic range (fundamental niche), and in this regard, we have to note that our analyses only explored the evolution of the realized niche and full laboratory experiments are necessary to confirm the fundamental niche of the species. Other alternatives to explain the lack of PNC in the bioclimatic

variables linked with temperature are probably related with the non-detection of microclimatic conditions by a scale problem of grain size—this condition could be more important for these species as the number of restriction hours (in refuges to avoid overheating) during the reproductive season remains <4 (*Sinervo et al., 2010*). This is as long as Bio15 and Bio19 remain within certain limits. For example, despite having preferred temperatures similar to other species of the group (*Sinervo et al., 2010*; *Martínez-Méndez, Mejía & Méndez-De La Cruz, 2015*), *S. serrifer* occurs in different habitats ranging from cold highlands in Chiapas and Guatemala to warm lowlands in the Yucatan peninsula; however, the latter area possesses certain tree species, sinkholes, and artificial refuges, such as walls and rock fences, which provide suitable thermal conditions to spend night and hours of restriction (*Martínez-Méndez, Mejía & Méndez-De La Cruz, 2015*). Thus, we believe that refuge microclimates and thermoregulatory behavior could permit this species to explore beyond typical montane sites and contribute to the lack of PNC detection in bioclimatic variables linked to temperature. Nevertheless, the multivariate phylogenetic analysis was able to detect a certain recent tendency to dry ambient for PC3 in the clade sustaining *S. serrifer* as the support for Pagel's delta ($\delta = 2.9$) indicated. In this sense, extensive ecophysiological, phylogeographic, and thermal ecology studies on the species of the group remain necessary to determine their fundamental niche and thermal requirements, and to measure the effect of biotic interactions and historical factors in its distribution.

In conclusion, our results indicate that the evolution of the *S. torquatus* group involved both niche divergence for the majority of bioclimatic variables and for PC2 and PC3, and evidence of PNC for Precipitation Seasonality (Bio15), Precipitation of Coldest Quarter (Bio19), and for PC1 scores, with only the latter potentially being linked to viviparity. This partially supports the prediction that viviparous species exhibit speciation via climatic niche conservatism, with similar niches being exhibited in allopatric sister species (*Lambert & Wiens, 2013*). Our data are strongly consistent with an allopatric speciation involving some level of divergent selection, where the heterogeneous environments of mountain ranges promote observed niche divergence with no similar niches between the majorities of sister species in this group (*Pyron & Burbrink, 2010*; *Pyron et al., 2015*). We believe that the availability of new climatically heterogeneous territories, the subsequent filling of that new environmental niche, and posterior cycles of isolation likely occurring during orogenic and glacial periods could account for the pattern we observed. Nevertheless, as previously mentioned, the physiological requirements and refuge use of this species must be evaluated in order to elucidate the most accurate pathway for niche evolution in the group.

## ACKNOWLEDGEMENTS

We thank Comisión Nacional para el Conocimiento y Uso de la Biodiversidad (CONABIO) for providing access to databases of occurrences or Mexican reptiles. We thank the support given by the Instituto Politécnico Nacional, particularly to the Escuela Nacional de Ciencias Biológicas.

### Funding
The authors received no funding for this work.

### Competing Interests
The authors declare that they have no competing interests.

### Author Contributions
- Norberto Martínez-Méndez conceived and designed the experiments, performed the experiments, analyzed the data, contributed reagents/materials/analysis tools, prepared figures and/or tables, authored or reviewed drafts of the paper, approved the final draft.
- Omar Mejía contributed reagents/materials/analysis tools, prepared figures and/or tables, authored or reviewed drafts of the paper, approved the final draft.
- Jorge Ortega contributed reagents/materials/analysis tools, prepared figures and/or tables, authored or reviewed drafts of the paper, approved the final draft.
- Fausto Méndez-de la Cruz prepared figures and/or tables, authored or reviewed drafts of the paper, approved the final draft.

### Data Availability
   Open Science Framework (OSF): https://osf.io/zn2eg/files/

### Supplemental Information
Supplemental information for this article can be found online at http://dx.doi.org/10.7717/peerj.6192#supplemental-information.

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
