# Peer review of "Climatic niche evolution in the viviparous Sceloporus torquatus group (Squamata: Phrynosomatidae)"

_PeerJ, doi:10.7717/peerj.6192_

## Round 0.1 · original submission · Major Revisions

All the reviewers felt that this paper had potential to provide a novel and meaningful analysis on an interesting topic. However, all had substantial concerns. In particular, I encorage you to address the following issues:

1) Writing. While comments on writing quality were variable, reviewer #3 gives many good suggestions to improve the language.

2) Framing. Reviewers felt that the paper could have been better framed and suggested focusing on the cold-climate hypothesis and how PS and PNC can help illuminate the evolutionary dynamics. In addition, take care that your paper is written for a broader audience. Watch out for excessive jargon and also realize that people who are not herpetologists are not going to be familiar with the various species groups you refer to (or even what a "group" means phylogenetically). It would be useful to introduce information about the larger sceloperous group and which species are or not viviperous - the story about how the groups you focus on were moving from cold to warmer climates really doesn't come clear in the paper.

3) Previous literature. Reviewer 1 highlighted a potential glaring omission of a relevant paper that explored the same hypothesis for the larger group. Make sure that you include that paper in your revised background and address how your results fit in and add new information.

4) Methodology. Reviewers 2 and 3 in particular had many methodological suggestions, some of which require reanalysis. Please carefully consider their suggestions and address reasons if you decide not to follow their suggestions.

5) All reviewers had some concerns with interpretations of your results. Please carefully address those and either elaborate on your logic or modify as necessary.

Reviewer 1 ·

Basic reporting

This is a well-written manuscript that addresses climatic niche evolution in lizards.

Tables:

Please round decimal places, it will make the tables more readable. Table 3 could be moved to the SM.

Figures:

Generally, there are too many figures, and some of them will not be readable in the final PDF. Please choose figures wisely and move the rest to the SM. Figure 1 could be combined with Fig. 2b. Captions for Figure 3-4 could explain what these results mean. Figure 5-6 will be very small and difficult to understand.

Supplementary material:

I appreciate the inclusion of the raw data for review and also for the reader eventually. However, in their current state, some of the supplementary material is impossible to decipher. The locality PDF for example, is split up in such a way that cannot be read. I'm not sure if this is PeerJ converting it a PDF, but if PeerJ does not accept excel files as SM, please include in another data repository.

Experimental design

Most of the analyses appear to be sufficient to address their hypotheses and are well explained and rigorous. The paper could be improved by better linking their analyses and results to the cold climate hypothesis and viviparity generally.

Validity of the findings

The discussion section poorly links back to introduction, and seems to restate the results too often, where readers will be interested in how these results support or reject the cold climate hypothesis. In addition, the discussion can better include the background research available for the topic. Lambert & Wiens (2013) is a glaring omission in the discussion, which tests this very same hypothesis in the larger group of Sceloporus lizards. In particular, Lambert & Wiens (2013) directly address this species group, pointing out that they represent an invasion of warmer climates from cooler climates. Therefore, phylogenetic history is a very likely factor that explains the origination and retention of viviparity, while at the same time explaining why there is niche divergence here. I think this is a very interesting and important point missing from the discussion.

Additional comments

Line Item comments:

Line 53-57: This paragraph would be better after the next paragraph (Line ~82).

Line 56-57: Confusing wording; aren't all species viviparous in this group?

Line 82-88: Not clear what the purpose is of each of these specific objectives and they relate to your hypothesis.

Line 95: how did you know the tossed-out records were from misidentifications?

Line 188-196: Why not use BEAST to estimate divergence dates? The dates are quite different from previous studies (e.g. Lambert & Wiens 2013), where the age of the group is half in comparison.

Line 388-389: This makes sense, but the interesting question is how does this phylogenetic history lead to the patterns we see today? Colonization of warmer climates from cooler climates would explain all the niche divergence.

Line 412: What about mean temperature of the warmest quarter (Bio10)? This group of lizards lays their eggs during the warmest months of the year (Jones and Lovich 2009)?

Line 563: Not all climatic variables (i.e. tolerances) would be expected to be constrained by viviparity, only variables relevant to the egg laying season (e.g. Bio10 and Bio5). I think it's not entirely relevant to consider all of the variables, when only a few might be important and conserved.

Reviewer 2 ·

Basic reporting

The authors of this study tested for niche conservatism (PNC) on climate occupation in a group of viviparous lizards of the Sceloporus torquatus group. They aim to test if this life history trait (viviparous) might represent an ecological constraint on the occupation of different climate conditions. Although I found their data, sampling and evolutionary questions interesting I think the manuscript needs revision.

I would recommend the authors to re-design some of the analysis they performed using the data they already have, in order to make the design more appropriate to what I understand are the main aims of their study. I found is not clear to me in their manuscript how is that viviparity should represent and evolutionary constraint to occupy cold habitats expecting to find PNC (see comments below for lines 35 - 43). Additionally, the study is sometimes hard to follow because it oscillates between the tempo and mode of diversification and the ecological consequences of viviparity in this group of lizards. These instances should be revised throughout in order to clarify the main objectives of the study and explain what the expected patterns are. Bellow I summarize some comments that I hope can help them to improve their manuscript.

Experimental design

It’s not clear to me why they are expecting that viviparous species should be evolutionary constrained on the climatic niches the species with this trait occupy and thus show PNC. If viviparity is supposed to avoid egg mortality linked to cold temperatures observed in OVIPAROUS species, why is that viviparous should be ecologically constrained? Besides that, these changes on climatic occupation between viviparous and oviparous species might only be reflected in the breeding season

Multivariate estimates of climatic niche occupation might give them more approximate insights about the diversity of ecological conditions (in this case the climate space of the niche) the species occupy and the historical evolution of this occupation. So I would recommend multivariate analyses to integrate different dimensions of the species climatic niches rather than making analyses based on single predictor variables

Validity of the findings

They stated that “there is no evidence that climatic niche differentiation was the main factor in the diversification of the studied group” does this contradict the fact that they did not found evidence for PNC based on their results of lack of PS?

The link of orogenic and glacial periods and the diversification of the group is not well explained. Although this might be discussed as a potential hypothesis I think it should not be included as a conclusion of the study since is not directly linked to their aims and their methods are not the most appropriate to test this hypothesis

Additional comments

I think they should introduce the study explaining the evolution of the viviparous strategy in lizards. Then, discuss the possible evolutionary and ecological consequences of this life history trait and mention what the other studies have found so far. This will introduce the reader to their main question of the study and clarify what this new study will bring to the discussion on the evolution of viviparity on lizards. This would stands as a better introduction rather than discussing ecological niches and PNC as an introduction.

2 do they refer by “actual” to the current distribution?

6 the authors should clarify that their concept of ecological niche is based on abiotic factors, but not that they refer to it through the text as estimates of biotic and abiotic interactions. Clarify

7 which are those ecological components?

9 what kind of speciation? Sympatric, ecological?

2 -15 is a little confusing, I would recommend to introduce their study with the evolution of viviparity and after that explain that PNC is the predicted pattern we should expect to observe when some life history trait represents an evolutionary constraint (in case viviparity really should represent an evolutionary constraint)

40 such as which life history traits?

56 -59 the three aims they mention here are very inter related to one another and to the ones stated at lines 86 – 88 (see comment below). Remove or try to be less redundant

74 mention how many species are present in these groups (grammicus and megalepidurus). The authors also should mention in what direction excluding these groups are expected to bias their results. If these groups habit on very different climatic habitats than the included group, excluding them should bias their results toward PNC making they inferences more robust (i.e. there is lack of PS even when excluding groups ecologically different than the main species group)

86 - 88 the aims 3,4,5, are very related and they might not stand alone as single aims of the study.

97 occurrence points should be removed at the same spatial scale than the environmental layers. A 0.5 km cut does not guarantee that all the points falling in the same pixel were removed because the resolution of the layers they used is ~ 1km x 1km

108 The recommend output for phyloclim analysis is the raw output from MaxEnt niche models, why the cloglog output? Which was the method used for MaxEnt replicates: bootstrap, cross-validation? They should also mention which version of MaxEnt they used or if these analyses were performed in some R package

119 how they choose 11 layers if they previously selected 10 layers according to their MaxEnt results and then remove the most correlated?

128 check previous comment on raw outputs

130 – 138 they might only mention that they performed partial ROC to test their models. Since this is more robust than AUC with pseudo absences (the default performance test implemented by MaxEnt)

234 – 242 the statistical support of the models of evolution could be simplified using likelihood ratio test instead of labeling models as more or less equivalent, distinguishable, etc.

239 242 why all the variables if these are not the most important on MaxEnt analyses

254 a positive and signifficant correlation is the pattern expected if there is PNC (species tend to have more niche overlap as they become more distantly related) and signifficant and negative correlation is the pattern expected if there is not PNC (species tend to have less overlap as they become more distantly related). But not that both signifficant correlations are evidence of PNC as stated in line 254

251 – 263 with which variables they performed these analyses?

266, 269, 271 remove niche tolerance or replace by niche occupation

313 - 314 the authors should also mention what is the general support for the node within these clades to give the reader an idea of how well resolved and supported their phylogeny is throughout

353 remove fails to detect PS since this test is not the most appropriate to do that

358 PNO profiles seem to be not that heterogeneous and they show many overlap among species
386 remove “other statistical bias” or be more specific about which these biases are

481 482 If they want to test if the species occupy different habitats it might be more appropriate to integrate multivariate analysis (for instance testing for phylogenetic signal on PC score values of multivariate PC axes). Since the pattern could be only associated with the breeding season but not in all the variation of climate occupation the species have throughout the year

491 change 20015 for 2015

493 this might not be reflected in the scale of their variables

498 how low niche overlap is evidence of niche conservatism? Isn’t this a contradiction?

524 525 absence of PS is evidence that climatic niche differentiation was not the main factor in the diversification of the group?

527 remove tolerance and change with occupation

529 same here

Reviewer 3 ·

Basic reporting

Grammar and english used in this ms made it difficult to read, and needs substantial work and refining. Some paragraphs are normal length, others are overly long. Past and present tense is confused throughout. Specific scientific listing of species (i.e., using the full genus name at first mention and at the beginning of sentences, followed by abbreviation thereafter) is also not followed throughout. Most background is provided, but there are substantial sections of the introduction that are not necessarily related to the study. References seem appropriate. Some analyses are inconclusive and should probably be excluded. Otherwise structure seems good. I do not necessarily consider the conclusions drawn here to fit with the results.

Experimental design

The data here are largely from published sources, but ideas are original and interesting. The questions asked here are relevant and meaningful and if undertaken correctly would fill a research gap. In my opinion, the data are overly speculated upon and do not provide as concrete evidence for the conclusions drawn as is written - and nee to be toned down substantially. There are often details missing from the methodology that preclude a complete understanding of what was done, and would certainly preclude replication of this study by others.

Validity of the findings

I think the results here are overstated in the conclusions drawn - and there is substantial speculation regarding the conclusiveness of these results. Sample sizes are relatively low for power in the main tests used herein to draw conclusions, others of interest (i.e., DTT results) are not examined at all despite interesting patterns emerging from the data.

Additional comments

Specific comments: I am providing substantial comments here in order to assist the author in refining the ms to be suitable for publication.

Abstract:
Ln 1-2: Correct to "...explanation for the evolution of viviparity in reptiles. This hypothesis maintains...."
Ln 7: Correct to "...to test whether their diversification is ...."
Ln 8-10: Correct to "...We evaluated phylogenetic signal and trait evolution, along with reconstructions of ancestral climate tolerances, and did not find evidence in support of PNC in the ecological niche of the species group."
Ln 11-12: Correct to "...; we only found evidence of PNC in ...."
Ln 13: Correct to ..."Analyses of relative trait disparity through time..."
Ln 13-16: First it is stated that there is evidence for divergence associated with orogenic cycles, and no PNC - but then it is stated that this is not due to niche differentiation. These statements somewhat contradict each other - and I would argue that there is evidence herein that there is niche differentiation (i.e., the absence of conservatism)

Background:
Lns 2-43: this should be split into at least 3 paragraphs - not as a single paragraph
Ln 2: What does "actual" distribution mean? Are you referring to the realized or potential niche of the species?
Ln 8: Correct to "...are important for the speciation...". Also note: it is now commonly believed that divergent natural selection alone cannot lead to speciation without cessation of gene flow through allopatry to assortative mating. See (Cotto & Servedio 2017 AmNat 190 (5), 680-693; Servedio & Boughman 2017 Annual Review of Ecology, Evolution, and Systematics 48, 85-109 as a start)
Ln 14: Correct to "...that limit adaptation to climatic..."
Ln 18: Phylogenetic signal in climatic traits?
Ln 19: Correct to "...random from a phylogenetic tree..."
Ln 21: Correct to "...have highlighted theoretical problems with the PNC concept and practical..."
Ln 24-25: Correct to "...questions are developed..."
Lns24-32: These arguments are confusing. I urge a rewrite for clarification of arguments.
Ln 33: The author suggests a topic is "barely known" but cites a reference from 1985. Have there been no recent developments in this field or at least a more recent paper citing the lack of this knowledge?
Ln34: Correct to "...interaction of these constraints..."
Ln 35: Correct to "..In this regard, viviparity..."
Ln 39-40: Which life history traits specifically?
Ln 44 Correct to "Viviparity among squamata..."
Ln 45 add a comma after the Blackburn references
Ln 46 Correct to "...viviparity in vertebrates..."
Ln 47: Correct to "...evolutionary studies regarding the relationship between niche evolution..."
Ln 48: Correct to "...Sceloporus. Sceloporus species are widely distributed in North America, contain..."
Ln 50: delete "for which there is"
Ln 51: Correct to "...species is available along with a comprehensive occurrence..."
Ln 53-54: Correct to "Given the hypothesis that the development of viviparity in reptiles is linked with the occupation of cool climates, this could lead to constraint in niche evolution among viviparous ..."
Ln 55: delete "organisms" after "model"
Ln 56: Correct to "...assess if niche evolution is phylogenetically constrained among viviparous.."
Ln 57-58: Correct to "...tolerances among species relative to their phylogenetic relationships fits with the PNC hypothesis; and 3) test whether the most.."
Ln 59: Are these the most important variables determining the extent of distributions? Please be more specific here.
Lns60-81: This paragraph should be either put in the methods, in a more concise format, for justifying species selection - or simply removed as most of the information presented here seems irrelevant to the paper.
Lns82-85: Correct to "...used phyloclimatic analyses that use occurrence data and bioclimatic variables to: 1) evaluate phylogenetic signal of bioclimatic variables comprising species niches,..."
Ln 86: Please define what variables are most important for?
Ln 88: Correct to "...calculate ecological niche disparity..."

Materials & Methods:
Ln 96: Correct to "...and mistakes in coordinates (i.e., points on the sea). To minimize..."
Ln 99: Correct to "...arc seconds). For environmental layers, we used bioclim layers at 30 arc seconds..."
Ln 100: Correct to "...1km) including monthly..." Delete s from "temperatures"
Ln 101: Correct to "...precipitation variables from the WorldClim.....(http://wwwwroldclim.org). We also used annual..."
Ln 102: Correct available to "variable" and put parentheses around the website.
Ln 104: Correct to "...clipped to limits of the distribution of all species in the group combined." This would allow for overprojection of species distributions, expecially for those with much narrower distributions relative to other species, impacting the validity of the results. A more appropriate method is to clip the layers to each species distribution and undertake MaxEnt analyses on each of these data separately. This will avoid overprojection problem.
Ln 116: Correct to "...relationships..."
Ln 119: Were those layers based on those that were important for all species or most important across species? How was this determined?
Ln140: Correct "achieved" to "performed"
Lns 143-155: This is the first time that it is mentioned that both the torquatus and poinsettii groups are being used. They are also listed as "former" groups in some places later on. This is neither clarified or explained and I feel should be mentioned before now in the text.
Ln 143: Correct to "...tree of the Sceloporus..."
Ln 144: Correct to "...group that resolved some..."
Ln 145: Delete being and add a comma after "studies"
Ln 146: make a new sentence after the references and change "they to Leache et al. (2016)
Ln 147: add a comma after the first "species"
Ln 148: Change "like" to "as the"
Ln 149-150: Correct to "...(2016), correpsinds to S. cyanogenys according to Martinez-Mendez de la Cruz (2007), therefore should not have affinity with S. serrifer..."
Lns151-155: I would be hesitant to take this statement as conclusive evidence of a misID given the stated paper is based on mtDNA alone and using different specimens (i.e., UWBM 6636 is listed in Leache et al (2016) using high-throughput sequencing while UTA-R-24004 is used for the Martinez-Mendez de la Cruz (2007) paper based on a few mtDNA genes). Are there morphological reasons to believe that the species' can easily be misIDed? What are the distributions of the two species'? Do they both occur in this region? These are relevant pieces of information that are not clarified here and make this statement as currently written inconclusive.
Ln 156: Correct to "...S. torquatus group"
Ln 157: Correct to "...we used sequences from four..."
Ln 160: add "the" before "grammicus", delete the "the" and change to "outgroup"
Ln 163: "outgroup"
Ln 163-164: Leache et al. (2016) actually finds 100% support for reciprocal monophyly of these groups - not "problems" as stated here. Therefore the provided reference here contradicts the authors statement.
Lns165-173: PartitionFinder would likely be a more appropriate choice of program to assess models of molecular evolution and data partitions here given these are largely concatenated analyses.
Ln 173: Correct to "...Phylogenetic relationships of members of the torquatus..."
Ln 179-180: Correct to "...inclusive parameter-rich model for analyses."
Ln 185: What does "edition" mean in this context?
Ln 193: Correct to "...the toquatus and poinsettii species groups..."
Ln 194: Correct to "...S. serrifer..."
Ln 199: Correct to "...Despite criticisms of the methods used to estimate PNC following..."
Ln 203: Correct to "...(BM) mode of trait evolution,..."
Lns 205-209: this is confusing as written and should be refined.
Lns199-242: This should be broken into at least 2-3 paragraphs
Ln 212-213: Correct to "...with 1000 simulations, Pagel's.......using a maximum likelihood method."
Ln 223: Correct to "...to test for fit of the data to four alternative models of trait evolution: 1)..."
Ln 224: delete "as we pointed out models" and "the"
Ln 225: add "trait" before "evolution, and delete "of a trait"
Lns 223-232: Disentangling a listing of the four models tests and their definitions would assist in making this text less confusing.
Ln232: Correct to "...of trait evolution was assessed using log likelihood values and..."
Ln 233: Correct to "...AICc was deemed the better..... Additionally, to clarify our ability to distinguish..."
Lns 231-239: It is not actually clear which of these you used to determine the best model. Also given your sample sizes it is not surprising that you in fact have little ability to distinguish among models for any of these variables. Bio15 is the only variable for which there is a clear model that can be distinguished and it fits a BM model of evolution. See comments below about assuming the results of these analyses are conclusive.
Ln 243-245: What data are you using here?
Ln 247: adaptive shifts?
Ln 253: Correct to "...method uses a..."
Ln 255: Correct to "...also be indicative of speciation.."
Ln 256: move parenthetical statement to after Warren's I
Ln 257: Correct to "...), both of which range from......"
Ln 258-259: Correct to "...that are likely incorrect, where there were significant differences between values of the I statistic relative to Schoener's D...". Also - what are these unrealistic assumptions specifically?
Ln 260 Correct to "...for further analyses,..." Also - if there are problems with one, why use both, not just the more accurate one?
Ln 267-268 Correct to "...relates suitability scores within species distributions (from MaxEnt analyses) to..."
Ln 269: Correct to "...of each species for..."
Ln 273: How were weighted means calculated?
Ln 282: Do you mean pairwise distances in weighted average PNO's? How was this calculated? If distances you are basically analyzing the evolution of similarity and divergence in climatic variables. This is ok - but should be clarified here and detailed in the explanation of the results relative to the hypothesis being tested.
Ln 284: Correct to "DTT is calculated..."
Ln 286: add "trait" before "evolution"
Ln 287: Correct to "...subclade was plotted..."
Ln 289 Correct to "...indicate trait disparity is distributed..."
Ln 290: Positive values for MDI only infer early trait evolution if they appear early in the evolutionary history of the group, consistent with an early burst model. Therefore, these two aspects - divergence between subclades, and early evolution - are being confused here.

Results:
Ln 299: change "of" to "for"
Ln 309: Correct to "Our reconstructed phylogeny..." and list specifically the figure being referred to.
Ln 324: What was the bootstrap value for the ML analyses?
Ln 327-Correct to "...(Bio15) had significant support for the PNC hypothesis (Table 1), with a moderate to weak PS and with variance..."
Ln 332 Change "is" to "was"
Ln 333: Change "are" to "was"
Ln 334: Delete "are" and add "models" after "equivalent"
Ln 335: Delete "of" and add "in support of"
Ln 336-337: Correct to "...This implies weak support for the PNC hypothesis for Bio19 using the ...."
Ln 339: change "are" to "is"
Ln 340: add "of trait evolution tested." after "models"
Ln 341: change "in" to "for"
Lns330-345: My interpretation of the data in Table 2 are that a BM model cannot be excluded as the best fit model in almost all of the variables (Except Bio19). The only variable to show conclusive support for one model (according to published criterion for distinguishing trait evolution models using AICc differences >2) here is Bio15 - which shows conclusively a BM model fits relative to others. Almost all of the discussion and interpretation of other results (i.e., that a BM model does not fit) is based on these analyses - which show no definitive conclusions. This is likely due to limited sample sizes in the current analyses - but this is not acknowledged until a small comment in the discussion - there should be acknowledgment of this problem before then. Therefore, there is way too much emphasis placed on the reliability of these results and they are vastly overstated in their importance and conclusiveness throughout the ms. This is simply over-inflating confidence in the conclusions and all language related to the reliability of these tests should be modified throughout the ms given these facts.
Ln 346: Change "are" to "were"
Ln 347: Correct to "...species, and within the toquatus and poinsettii clades. Similarly, only a few species paris show.."
Ln 348: Correct to S. cyanstitctus
Ln 349: Correct to "...these were sister taxa or closely-related species in our analyses, with...
Ln 353: add an s to "layer"
Ln 354-355: Correct to "...,excepting Bio15."
Ln 358: Which variables specifically?
Ln 361: Correct to "...peaks indicate that similar climatic tolerance between a few species was observed in..." Also, which species specifically?
Ln 362: change "are" to "was"
Ln 363: delete "the" before "Precipitation", change "has" to "had"
Ln 364: Correct to "...that was narrower than other bioclimatic layers, which is consistent with an OU model of trait evolution with a single optima, as was detected...". Also, here the OU model cannot be distinguished from a model where divergence increases towards the present (as indicated by a delta >1). The delta model arguably receives support from increased MDI toward the present in DTT plots for this variable - albeit not outside the CI of the null BM model. This would contradict this statement.
Ln 366: Correct to "...S. serrifer, which shows more extreme..."
Ln 368: Correct to "...plots detailing the history of climatic tolerance evolution..." Also - what you are looking at here is not climatic tolerance evolution if you have used the differences between species as outlined in the methods. You are looking at the history of similarity and divergence of climatic tolerance. If you used the weighted means of PNO's and not their differences this statement would be correct.
Ln 371-377: These statements need to be modified given the problems with over-interpretation of the conclusiveness of analyses showing models of trait evolution.
Ln 377: Correct to "S. serrifer"
Ln 379: More influence on their distribution?
Ln 389: Bio 9 and 15 don't appear different to me at all relative to the other analyses and nothing early on falls outside the 95% CI of the null BM model (grey shading in DTT plots)
Ln 390-392: I disagree with this interpretation as variables almost never go outside the 95% CI for most variables here. The exceptions are Bio5, Bio8, Bio9, and maybe Bio19. They all show increasing divergence among subclades towards the present (correction of time axes for Fig. 6 will make this clearer). This fits with the delta values presented in Table 2 - but which cannot be distinguished relative to other models for these species.These are the interesting results from this analysis and they are almost not even mentioned here. You cannot interpret values that do not exceed the confidence limits of the null model (grey shading) just because they are greater than the average values from the null (dotted lines).
Ln 395-408 - I find this section overly speculative. It should be modified or deleted, also "relative time" here means nothing. Authors would usually correct these values to the relative time calibrations on their tree if they are available.

Discussion:

The problems highlighted above persist in the discussion, I have refrained from editing this as I feel there are substantive enough problems highlighted above that the discussion will need to be completely rewritten.

---

## Round 0.2 · Major Revisions

Please pay close attention to the review and address the issues raised. There are two major issues - one is that the writing and framing of the paper are still not clear enough to properly evaluate your paper. It is unclear from the writing whether this is a test of the cold climate hypothesis, or this hypothesis is accepted and you are now trying to determine if this adaptation to cold climates (viviparity) is acting as an phylogenetic constraint. Note that the reviewer rightly points out that you never give a rationale for why this should be.

Also, and importantly, you need to more fully place these results into those found by Lambert and Wiens (2013) - you don't give enough information so that readers can make this assesment without them going to the original article. Also, your manuscript should be updated to account for the new paper by Ma et al. 2018.

Finally, the discussion and results are very difficult to follow. Basically, both the framing of the study and the writing (especially the discussion) need substantial work. Only then can it really be assessed in terms of publication.

Reviewer 1 ·

Basic reporting

A) The English and grammar could still use a large amount of work. The paper is challenging to follow at times, especially when there are paragraphs that exceed a page and a half in length (lines 437-483). The introduction has improved a great deal (but could still use better framing of their hypotheses) and I think most of the readability problems are found in the rest of the paper, especially the discussion, which is very difficult to follow. These areas (some pointed out below) need to be completely rewritten and condensed to their most important points.

B) Previously I suggested the authors discuss the cold-climate hypothesis more, and here they actually remove all mention of it from the main text. Oddly, the cold-climate hypothesis is mentioned several times in the abstract and that their study does not support it. The authors need to address this hypothesis in the introduction, why it is relevant to their study, and also how their study is testing this hypothesis or something different (which is not immediately clear). Additionally, a recent paper was published (below) that should be addressed here that may help the authors better add context to their study:

Ma L, Buckley LB, Huey RB, Du W‐G. A global test of the cold‐climate hypothesis for the evolution of viviparity of squamate reptiles. Global Ecol Biogeogr. 2018;27:279–289. https://doi.org/10.1111/geb.12730

C) Figures are improved and look readable now.

D) Authors should upload raw data to an online repository. They could use the center for open science (open science foundation) which is completely free, and they can provide private links for reviewers. https://osf.io

Experimental design

A) The rationale behind the study is not well explained. From what I could ascertain, the authors argue that viviparity should constrain the group to remain in cold climates because viviparity constrains other traits, so there might be some correlation in climate to these constrained traits. I do not think this is a strong argument. Viviparity presumable evolved in cold climates to deal with the lack of warmth for eggs in the external environment. However, it should be explained why this should act as a constraint in climatic variables after the trait has evolved. Are there studies that suggest that viviparity is disadvantageous in warmer climates?

B) What this study tests differently from Lambert & Wiens (2013) is not clear. Lambert & Wiens 2013 tested the cold-climate hypothesis on the same species group as this paper, and show that this species group represents an invasion of warmer climates from cooler climates, which already provides evidence against the hypothesis of PNC they aim to test here. The authors need to distinguish the two studies more clearly if they are different. If they are actually testing the same hypothesis, then this needs to be clear because PeerJ allows replication of prior studies, and different analyses are especially welcome. The true aim should not be as obscure as it is presented. The authors state their study is quite different than Lambert and Wiens 2013 but it appears quite similar, for example, Figure S6 here is basically showing the same thing as Lambert and Wiens 2013 Figure 1.

Validity of the findings

A) It is not clear how the conclusions link back to their original research question, as the authors waver between testing the cold-climate hypothesis (explicitly stated in the abstract) or whether viviparity constrains species to inhabit cold climates (stated in the introduction). If they are testing the cold-climate hypothesis, they should link this constraint back to it and why it would be a prediction of this hypothesis. If not, they should distinguish their hypothesis from the cold climate hypothesis, and explain both clearly in the introduction. I could envision arguments going both ways here.

B) More explanation for the lack of a positive result for PNC is needed. Lambert & Wiens 2013 suggest that phylogenetic history could explain retention of viviparity because viviparity evolved first in cold climates, but that cold climates are merely a prerequisite for viviparity to evolve, not a constraint as they show multiple colonizations of warmer climates by these viviparous species.

C) Why should we expect viviparity to act as a constraint for cold climates? Could this be reframed as a general expectation of PNC to cold climates? Since viviparous species occur in warmer climates, what previous research could be addressed to explain this?

Additional comments

This manuscript addresses climatic niche evolution in lizards and has improved since the last review in some ways, and has worsened in others. For this re-review, I also took into account what the other reviewers had suggested. Most of the analyses appear to be sufficient to address their hypotheses and the additional analyses are welcome, as as well as the requested changes. I have a few line item comments below:

Line item comments:

Line 43: Unclear who the “some authors” are. The cited paper goes into detail on the numerous ways to test for PNC, so it clearly not this paper you are referring to.

Lines 238: Why didn’t you test the white noise model? Support for a white noise model would support the hypothesis that there is no PNC. Otherwise the negative result from the other models could suggest many things, including a lack of power.

Lines 438-442: Lambert and Wiens 2013 does address whether viviparity constrains these species to cold-climates. They discuss in detail the paradox between so many viviparous species inhabiting warmer climates and suggest it is due to the trait first evolving in cold-climates and the species later dispersing to warmer climates.

Lines 437-483: Very long paragraph. Its also not entirely clear what the point of this paragraph is, as it just basically lists potential traits that could be related to climatic variables. I’m not entirely sure why its relevant.

Lines 498-500: This sentence is not clear.

Line 498, 500: Ravell is spelled “Revell”

Lines 541-578: Another long paragraph that is a page long.

---

## Round 0.3 · Minor Revisions

This manuscript was much improved! However, there are still some areas where increased clarity would help. Pay special attention to the reviewer's comments about the discussion.

In addition, for the non-specialist, I have a few suggestions that may help clarify the introduction.

First, I had to read into the details - but I come away with the sense that your overall framework seems to be to assume that niche conservatism causes sister species on mountaintops that evolved from a common ancestor to show similar niche requirements and phlogenetic constraints then keep them from expanding their range into lower altitudes. If this is a correct take, it would be useful to really come out and say this directly - but also potentially have a figure that shows the distribution of the species in the focal group and which ones are still viviperous (all?) - and their current distributions relative to altitude. That would really help me see your points more clearly.

Also, be careful about the way you set up the PNC framework - especially in lines 52-55 you seem to dismiss it entirely.

The discussion of Grinellian niches seems to come out of left field - remind people of the different niche definitions and also how this relates to the obvious environmental gradients from mountain-tops to sea-level.

Reviewer 2 ·

Basic reporting

I have reviewed the article “Climatic niche evolution in the viviparous Sceloporus torquatus group (Squamata: Phrynosomatidae)”. The study does not have methodological or sampling issues and the analyses are proper for their questions. The authors have followed most recommendations of reviewers and editor. Therefore, I would recommend this revised manuscript for publication. However, I still recommend the authors to review the English for a few mistakes I found (see some examples below in minor comments) and maybe work a little more to make a more readable and straight to the point manuscript, especially in the discussion section (see major comment below).

Experimental design

no comment

Validity of the findings

no comment

Additional comments

Major comments

The discussion is sometimes hard to follow. It tends to over discuss by comparing the results of the different analyses they performed but, in many cases, not making these discussions in terms of their predictions but in terms of how congruent are (or are not) the different approaches they used to test PNC and the theory behind this.

The authors should mention which phylogenetic tree they used for their comparative analyses. Was this the consensus? The dated tree? I couldn’t find this information in the text.


Minor comments

L7-8 It’s not clear what they meant by “using bioclimatic variables relevant to each viviparous group”. This is one of the justifications for their study but it was not clear to me, how the previous studies did not use relevant variables?

L65 the word “interactions” might be more appropriate than “variables”

L66-67 not clear what they meant by “the environmental space of non-interacting abiotic variables”

L64-68 Thus what definition of niche are they using in their study? Not clear why they discussed the differences on these definitions but not connected them with their study/questions

L115 specify that the 1km x 1km it’s an approximation

L146 “Final Maxent analysis” not sure if it should be plural

L151-252 “The identification of best-fit model” sounds rare to me

L282 “I tended to yield higher values” are they referring to their own results? Or this is a generality of this statistic?

L286 “resampling of overlap matrix” I think it should be “resampling of the overlap matrix”

L298 similar than the latter for “weighted means of PNOs”

L318 “For total phylogeny” sounds rare

L511-513 how PC3 represents dry ambient? Because the highest loadings are related to precipitation variables?

L530 “the ecological differences that detected in PNO profiles” sounds rare, maybe “the ecological differences that were detected in the PNO profiles”?

L586-589 Clarify that the bioclimatic variables used are not able to detect these microclimatic differences because of a scale problem of grain size

L640 what do they mean by “PC3 evolution”

---

## Round 0.4 · accepted · Accept

Great job on the revisions. I think the article is much clearer and will be more approachable for a general audience.

#